# Frustrated endocytosis controls contractility-independent mechanotransduction at clathrin-coated structures

Francesco Baschieri [1], Stéphane Dayot[1,5], Nadia Elkhatib[1], Nathalie Ly[1], Anahi Capmany[2], Kristine Schauer[2], Timo Betz[3], Danijela Matic Vignjevic [2], Renaud Poincloux [4] & Guillaume Montagnac[1]

It is generally assumed that cells interrogate the mechanical properties of their environment by pushing and pulling on the extracellular matrix (ECM). For instance, acto-myosin-dependent contraction forces exerted at focal adhesions (FAs) allow the cell to actively probe substrate elasticity. Here, we report that a subset of long-lived and flat clathrin-coated structures (CCSs), also termed plaques, are contractility-independent mechanosensitive signaling platforms. We observed that plaques assemble in response to increasing substrate rigidity and that this is independent of FAs, actin and myosin-II activity. We show that plaque assembly depends on αvβ5 integrin, and is a consequence of frustrated endocytosis whereby αvβ5 tightly engaged with the stiff substrate locally stalls CCS dynamics. We also report that plaques serve as platforms for receptor-dependent signaling and are required for increased Erk activation and cell proliferation on stiff environments. We conclude that CCSs are mechanotransduction structures that sense substrate rigidity independently of cell contractility.

[1] Inserm U1170, Gustave Roussy Institute, Université Paris-Saclay, Villejuif, France. [2] Institut Curie, CNRS UMR144, PSL Research University, Centre Universitaire, Paris, France. [3] Institute of Cell Biology, Center of Molecular Biology of Inflammation, Cells-in-Motion Cluster of Excellence, University of Münster, Münster, Germany. [4] Institut de Pharmacologie et Biologie Structurale, IPBS, Université de Toulouse, CNRS, UPS, Toulouse, France. [5] Present address: Institut Curie, Inserm U830, PSL Research University, Centre Universitaire, Paris, France. Correspondence and requests for materials should be addressed to F.B. (email: francesco.baschieri@gustaveroussy.fr) or to G.M. (email: guillaume.montagnac@gustaveroussy.fr)

Cells constantly probe the extracellular milieu in order to adapt to the changing conditions of the environment. Besides chemical signals sensed by specific receptors, cells also respond to mechanical stimuli with important consequences for cell migration, proliferation and differentiation[1–3]. It is generally accepted that cells probe mechanical features of the microenvironment by applying forces on it[4–6]. Contractile forces generated by the acto-myosin network and transmitted to the substrate at integrin-rich cell adhesions endow these adhesions to grow and mature into focal adhesions (FAs), in a matrix rigidity-dependent manner[7,8]. In turn, FAs maturation has profound consequences for the cell as it modulates signaling pathways regulating migration, survival and proliferation. Clathrin-coated structures (CCSs) are mostly described to control the uptake of cell-surface receptors, including some integrins. However, it is now clear that in some conditions, CCSs can also serve as integrin-dependent adhesion structures[9]. Many cell types, including HeLa cells, display two distinct types of CCSs: canonical, dynamic clathrin-coated pits (CCPs) and long-lived, large and flat clathrin lattices also called plaques. Although plaques have been widely described and shown to be enriched in signaling receptors and integrins[10–12], it is still not clear how they form and what is their function. CCSs have mostly been studied in cells growing on glass which is an extremely stiff substrate. A whole range of tissue rigidity is encountered in vivo with some tissues being very soft (Young's modulus, $E \approx 0.1$ kPa) like the brain or fat tissues, while some other are stiffer like muscles ($\approx 30$ kPa)[13]. Here, we set out to investigate CCSs dynamics on substrates of controlled elasticity. We report that clathrin-coated plaques assemble as a consequence of increasing substrate rigidity. Surprisingly, plaque formation on stiff environments is independent of cell contractility but is the consequence of a frustrated endocytosis process whereby αvβ5-integrin prevents CCSs budding by anchoring the structure to the substrate. We further report that receptor clustering at clathrin-coated plaques potentiates intracellular signaling and increases cell proliferation. In summary, we propose that clathrin-coated plaques are mechanosensitive structures instructing the cell about the rigidity of its environment.

## Results

**Clathrin-coated plaques are sensitive to substrate rigidity**. When HeLa cells were grown on collagen-coated glass, ventral plasma membrane CCSs marked with the α-adaptin subunit of the clathrin adaptor AP-2 appeared as a mix of dot-like, diffraction-limited structures corresponding to CCPs, and large, heterogeneous structures corresponding to plaques, as previously reported[11,12,14] (Fig. 1a). Strikingly, cells seeded on soft (0.1 kPa) collagen-coated polyacrylamide gels only showed dot-like CCSs suggesting that plaques cannot form in these conditions (Fig. 1a). Similar results were obtained with cells cultured on 5 kPa gels (Fig. 1a). However, cells seeded on 31 kPa gels showed a mix of diffraction-limited CCPs and larger structures potentially corresponding to plaques (Fig. 1a). Super-resolution STED microscopy analyses further confirmed the presence of many large CCSs in cells grown on glass or on 31 kPa gels while only dot-like structures were detected on 0.1 and 5 kPa gels (Supplementary Fig. 1a). Scanning electron microscopy analyses of unroofed cells confirmed the presence of large, flat clathrin-coated plaques at the adherent plasma membrane of cells cultured on glass or on 31 kPa gels (Supplementary Fig. 1b). Importantly, such large and flat clathrin lattices were mostly absent in cells seeded on 0.1 or 5 kPa gels (Supplementary Fig. 1b). We next performed live cell imaging of genome-edited HeLa cells engineered to express GFP-tagged, endogenous μ2-adaptin subunit of AP-2. Many CCSs

were large and long-lived when cells were grown on glass, reflecting the mostly static nature of clathrin-coated plaques (Fig. 1b, c, Supplementary Fig. 2a and Supplementary Movie 1). Similar results were obtained when cells were seeded on 31 kPa gels (Fig. 1b, c, Supplementary Fig. 2a and Supplementary Movie 1). However, the proportion of long-lived CCSs dramatically dropped in HeLa cells cultured on softer gels (0.1 and 5 kPa; Fig. 1b, c, Supplementary Fig. 2a and Supplementary Movie 1). Similar to HeLa cells, HepG2 and Caco-2 cell lines harbored both dynamic CCPs and large, static CCSs when cultured on glass or on 31 kPa gels while only short-lived CCPs were detectable on softer gels (Supplementary Fig. 2b–e). However, we did not measure any elasticity-dependent modulation of CCSs dynamics in MDA-MB-231 cell line that was previously reported to only harbor CCPs[15] (Supplementary Fig. 2f, g). Together, our results demonstrate that clathrin-coated plaques are mechanosensitive structures that some cell types assemble in response to stiff substrates.

Substrate rigidity sensing is described as an active process involving actin-dependent contraction forces transmitted to the ECM at FAs[4–6]. Thus, we tested whether acto-myosin-generated forces and FAs are required for plaques formation. Inhibiting myosin-II activity or actin polymerization with blebbistatin or cytochalasin D respectively, did not prevent the formation of large and static CCSs in HeLa cells grown on glass (Fig. 1d, e) and large, flat clathrin lattices were still detected by scanning electron microscopy in cells treated with blebbistatin (Fig. 1f). In addition, interfering with FAs assembly using Talin1-specific siRNAs did not prevent but rather increased the formation of long-lived, large CCSs (Supplementary Fig. 2h–j). These results demonstrate that plaque formation does not rely on cell contractility and suggest that plaques represent a new type of mechanosensitive structures.

**Integrin αvβ5 is required for plaque assembly**. In the classical mechanosensation model, cell-generated forces are transmitted through integrins binding the ECM at FAs. Although they assemble in a contractility-independent manner, plaques have been proposed to adhere to the substratum[16,17]. Integrin β5 was reported to be highly enriched at plaques[10,18]. β5 interacts with αv-integrin to form the high-affinity vitronectin receptor[19,20]. We observed that αvβ5 strongly colocalized with CCSs at the ventral plasma membrane of HeLa cells cultured on glass (Fig. 2a). Close inspection revealed that αvβ5 was highly enriched in plaques as compared to CCPs (Fig. 2a, b). The degree of association between αvβ5 and AP-2-marked CCSs significantly decreased on softer gels (Fig. 2c and Supplementary Fig. 3a). We next tested whether αvβ5 was required for plaque formation on hard surfaces. Knockdown of either αv or β5 integrins using specific siRNAs resulted in a complete loss of large and static CCSs (Fig. 2d–f, Supplementary Fig. 4a–d and Supplementary Movie 2). We did not observed any effect of β1 or β3 integrins depletion on CCS dynamics (Supplementary Fig. 4e, f). Inhibiting β5 in HepG2 cells also precluded the formation of large CCSs (Supplementary Fig. 4g). In addition, plaque formation was rescued by expressing a siRNA-resistant β5-encoding construct in β5-depleted HeLa cells (Fig. 2g). Of note, MDA-MB-231 cells that mostly harbor dynamic CCSs (Supplementary Fig. 2e, f) displayed drastically lower levels of αv and β5 integrins as compared to HeLa cells (Supplementary Fig. 4h). Overexpression of αv and β5 integrins in these cells resulted in a ~4-fold increase in long-lived CCSs (Supplementary Fig. 4i). Together, our results demonstrate that αvβ5-integrin is required for plaque formation on rigid environments.

Integrins are generally recruited at CCSs through binding to specific adaptors of the Phosphotyrosine Binding (PTB)

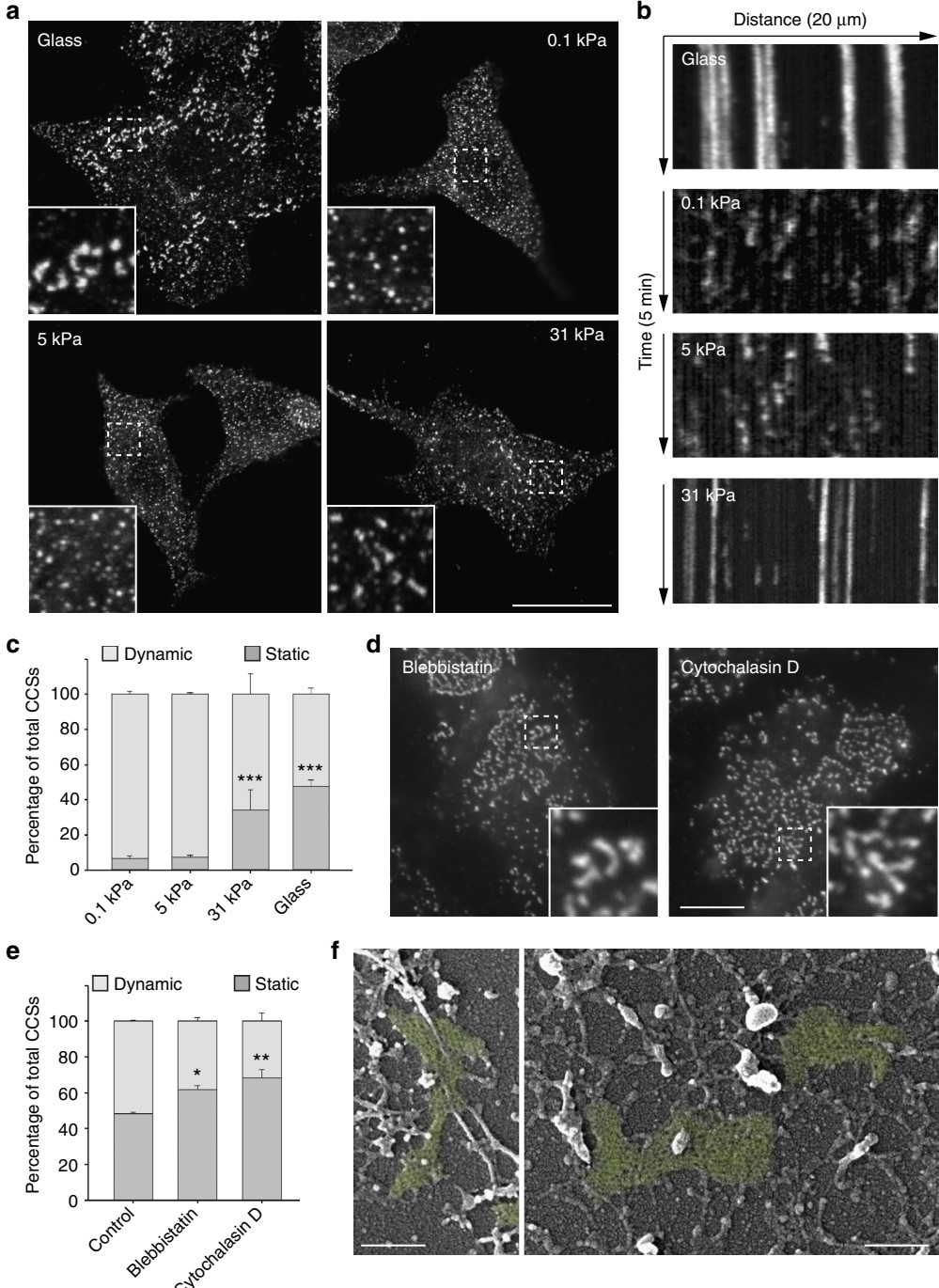

**Fig. 1** Clathrin-coated plaques are mechanosensitive structures. **a** HeLa cells were seeded on collagen-coated glass or polyacrylamide gels of indicated stiffness and fixed 24 h later before being stained for α-adaptin. Scale bar: 15 μm. Higher magnifications of boxed regions are shown. **b** Kymographs showing CCS dynamics in genome-edited HeLa cells expressing endogenous GFP-tagged μ2-adaptin seeded on the indicated collagen-coated substrate and imaged by spinning disk microscopy every 5 s for 5 min. **c** Quantification of the dynamics of CCSs observed as in **b** (***$P < 0.001$, as compared to 0.1 kPa condition, one-way analysis of variance—ANOVA. $n = 3$). **d** HeLa cells seeded on collagen-coated glass were treated with Blebbistatin or Cytochlalasin D as indicated for 30 min before being fixed and stained for α-adaptin. Scale bar: 10 μm. Higher magnifications of boxed regions are shown. **e** Quantification of the dynamics of CCSs observed as in **d** (**$P < 0.01$, *$P < 0.05$, ANOVA. $n = 3$). **f** EM micrographs of unroofed HeLa cells that were cultured on glass and treated for 30 min with blebbistatin before being fixed and processed. Clathrin-coated plaques are highlighted in green. Scale bar: 200 nm. All results are expressed as mean ± SD

domain-containing protein family. For instance, β5-integrin has been shown to interact with the clathrin adaptors Numb and Dab2[21]. Both Numb and Dab2 are present at plaques (Supplementary Fig. 5a) and inhibiting their expression using specific siRNAs led to a strong reduction of the size of CCSs as well as a moderate reduction of the proportion of long-lived structures (Supplementary Fig. 5b–e). In these experiments, we noticed a strong delocalization of αvβ5 to peripheral structures that were positive for the FA marker phospho-Focal Adhesion Kinase (FAK) (Supplementary Fig. 5f). A similar phenotype was

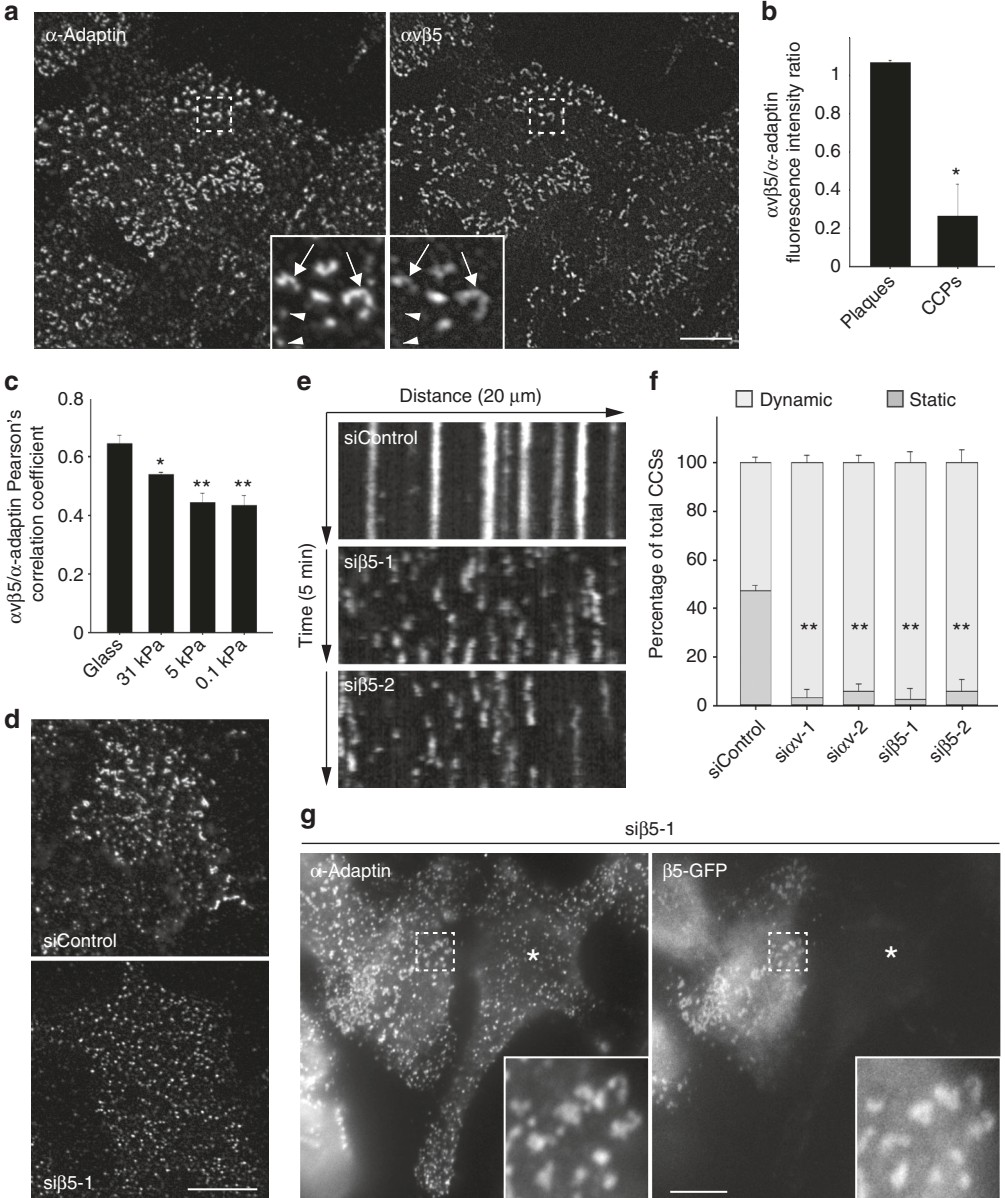

**Fig. 2** αvβv5 integrin localizes to plaques and is required for their assembly. **a** HeLa cells were seeded on collagen-coated glass and fixed 24 h later before being stained for α-adaptin and αvβ5-integrin. Scale bar: 10 μm. Higher magnifications of boxed regions are shown. Arrows point to clathrin-coated plaques; arrowheads point to CCPs. **b** Quantification of αvβ5 enrichment at plaques versus CCPs (*$P < 0.005$, two tailed Student's $t$-test. $n = 3$; 100 structures per experiment were counted.). **c** Quantification of colocalization (Pearson's coefficient) between αvβ5 and α-adaptin in cells cultured on the indicated collagen-coated substrate (*$P < 0.005$; **$P < 0.001$, one-way analysis of variance—ANOVA. $n = 3$). **d** HeLa cells treated with control (upper panel) or β5-specific (lower panel) siRNAs were seeded on collagen-coated glass and fixed 24 h later before being stained for α-adaptin. Scale bar: 15 μm. **e** Kymographs showing CCS dynamics in genome-edited HeLa cells treated with the indicated siRNA, seeded on collagen-coated glass, and imaged by spinning disk microscopy every 5 s for 5 min. **f** Quantification of the dynamics of CCSs observed as in **e** and treated with the indicated siRNAs (**$P < 0.001$, ANOVA. $n = 3$). **g** HeLa cells treated with β5-specific siRNAs were transfected with a siRNA-resistant β 5-GFP encoding construct and then fixed 24 h later before being stained for α-adaptin. Scale bar: 10 μm. The star marks a cell that is not transfected by β5-GFP. Higher magnifications of boxed regions are shown. All results are expressed as mean ± SD

observed upon inhibiting CCSs formation with AP-2 or CHC siRNAs (Supplementary Fig. 5f). This suggested that αvβ5 may shuttle to FAs in the absence of clathrin-coated plaques. αvβ5 is only poorly associated with FAs in control cells and, accordingly, β5-depletion did not modulate FAs number and size, nor the activation status of FAK (Supplementary Fig. 6a–e). However, preventing CCSs formation with AP-2 siRNAs resulted in larger FAs and this was dependent on β5-integrin expression (Supplementary Fig. 6f, g). Thus, in the absence of CCSs, αvβ5 shuttling to FAs induces an overgrowth of these adhesion structures.

Together, our data suggest that Dab2/Numb-dependent recruitment of αvβ5 at CCSs results in the formation of clathrin-coated plaques and that in the absence of plaques, αvβ5 is targeted to and modulates the dynamics of FAs.

**Plaque formation is a consequence of frustrated endocytosis.** We reasoned that αvβ5 engagement with the ECM may prevent CCSs budding, leading to the formation of plaques. The main αvβ5 ligand is vitronectin, an ECM component that is present in

the serum and that was described as the serum spreading factor allowing cells to adhere on glass but also binding a plethora of other ECM components, including collagen[22,23]. We observed an increased density of large and stable CCSs when the glass was first coated with vitronectin before allowing HeLa cells to adhere in the presence of complete medium (Supplementary Fig. 7a, b). However, vitronectin coating did not result in more static CCSs on the soft environment (Supplementary Fig. 7c, d). In addition, most large and long-lived CCSs were detected at cell/ECM contact areas rather than at non-adherent regions of the ventral plasma membrane when cells were cultured on ring-shaped, collagen-coated micropatterns (Supplementary Fig. 7e–g). These data suggest that local and strong engagement of αvβ5 with the ECM may be required for plaque assembly. To test this hypothesis, cells seeded on glass were incubated in the presence of trypsin. We reasoned that trypsin-mediated αvβ5 proteolysis may lead to the disassembly of clathrin-coated plaques. Indeed, we observed a fast dissolution of plaques upon trypsin treatment (Fig. 3a). Dot-like structures seemingly emanating from dissolving plaques were frequently observed before they abruptly vanished (Fig. 3a, arrows). These observations suggest that CCPs may bud in the vicinity of disassembling plaques, potentially participating in the dissolution of these structures. Along this line, we observed that auxillin bursts, a feature of budding CCPs[24], were significantly more frequent at disassembling plaques as compared to plaques of untreated cells (Fig. 3b, c and Supplementary Movie 3). Because trypsin treatment is not specific towards αvβ5, we next monitored plaque dynamics in cells treated with Cilengitide, a drug that binds the RGD-binding cleft of this integrin and outcompetes ECM ligands[25–27]. CCSs of HeLa cells grown on glass and treated with Cilengitide were highly dynamic as compared to untreated cells (Supplementary Fig. 8a, b and Supplementary Movie 4), demonstrating that direct binding of αvβ5 to the substrate is required to maintain plaques. Moreover, we observed a fast dissolution of plaques when monitoring CCS dynamics immediately after adding Cilengitide in the medium (Fig. 3d, Supplementary Fig. 8c and Supplementary Movie 5). Similar to experiments performed with trypsin, plaques seemingly disassembled into dot-like structures evoking CCPs (Fig. 3d). Thus, acute inhibition of αvβ5 binding to the substrate may allow the clathrin machinery to bud, leading to plaques dissolution. Accordingly, we observed that overexpressed, GFP-tagged β5-integrin accumulated into mCherry-tagged transferrin receptor (TfR)-positive vesicles after but not before treating cells with Cilengitide (Supplementary Fig. 8d). This demonstrates that the integrin was internalized upon adding the drug, in agreement with our hypothesis.

Based on a theoretical model, it has been proposed that integrin adhesion strength is reduced on soft environments because local elastic recoil upon stochastic integrin/ligand unbinding would lower chances to rebind, independently of forces[28,29]. Such a mechanism may explain why plaques only assemble on relatively stiff but not on more elastic substrates. We observed that the kinetics of plaque disassembly upon Cilengitide treatment were directly correlated with the stiffness of the substrate as plaques disassembled faster on softer environments (Supplementary Fig. 8c). Because Cilengitide acts by outcompeting integrin ligands, these results suggest that αvβ5 adhesion strength is reduced on soft substrates.

Our data suggest that plaque assembly results from a frustrated endocytosis process whereby strong receptor-mediated anchoring to the stiff substrate prevents CCSs budding. To further test this hypothesis, we made use of a construct encoding the transferrin receptor (TfR) carrying a mCherry tag in its extracellular domain. β5-integrin-depleted HeLa cells expressing or not TfR-mCherry were plated on anti-mCherry antibody-coated glass. We reasoned

that immobilizing the TfR, which has a strong affinity for AP-2, may stall endocytosis and result in plaque formation. Indeed, while CCSs of β5-depleted cells were dot-liked and short-lived on the anti-mCherry antibodies-coated glass, cells expressing TfR-mCherry displayed many large and long-lived structures (Fig. 3e–g and Supplementary Movie 6). We further controlled that TfR-mCherry was present at CCSs in these conditions (Supplementary Fig. 8e). Of note, TfR-mCherry-expressing cells seeded on a coverslip functionalized with an irrelevant antibody where not able to assemble plaques upon β5-depletion (Fig. 3g). In addition, immobilizing the TfR on soft gels resulted in the accumulation of long-lived and large CCSs, although to a lesser extend as compared to the glass condition (Supplementary Fig. 8f). Thus, strong receptor engagement with the substrate is sufficient to prevent CCSs budding and to lead to plaque formation. Overall, our results demonstrate that plaques assemble as a consequence of frustrated endocytosis.

**Plaques are signaling platforms for different receptors**. It has been proposed that plaques are a hub for the recruitment/sorting of many signaling receptors[11,12,30]. Prolonged CCS lifetime has been associated with sustained signaling and CCSs have been envisaged as signaling platforms, independently of their role in endocytosis[31]. Thus, long-lived clathrin-coated plaques may modulate signaling pathways. In agreement with this hypothesis, we observed that the inhibition of plaque formation in β5-integrin-depleted cells cultured on glass resulted in a reduced steady-state signaling in the Erk pathway (Fig. 4a, b). However, β5 knockdown did not modulate Erk activity on a soft gel that do not allow plaque formation (Fig. 4c, d). This suggested that plaque assembly, but not simply β5 expression, is required to tune Erk activity. Surprisingly, AP-2 or clathrin heavy chain (CHC) knockdown did not modulate Erk activation status on a rigid surface (Supplementary Fig. 9a). Because in these latter conditions, αvβ5 shuttling to FAs leads to the enlargement of these structures (Supplementary Fig. 6f, g) and because FAs are also known to signal in the Erk pathway[32], we aimed at testing whether FAs could mask a potential role for plaques in regulating Erk activity. We first observed that AP-2 or CHC knockdown induced a strong accumulation of phosphotyrosines at the enlarged FAs (Supplementary Fig. 9b–d), in a β5-integrin-dependent manner (Supplementary Fig. 6f–g). In agreement with our hypothesis, these results suggest that the signaling activity of FAs is increased in the absence of CCSs. In addition, while the inhibition of FA formation in Talin1-depleted cells only slightly inhibited steady-state Erk activity, co-knockdown of Talin1 and AP-2 or CHC strongly reduced it (Supplementary Fig. 9e, f). Together, our results suggest that Erk activity is mostly controlled by CCSs in HeLa cells but that in the absence of CCSs, αvβ5-mediated FA enlargement maintains steady-state Erk activity.

To test more directly whether clathrin-coated plaques regulate Erk signaling, we took advantage of the possibility to rescue plaque formation in β5-depleted cells with the antibody-coating system described above. In these experimental conditions, cells are seeded on a layer of antibodies deposited onto a Poly-L-lysine coated glass. Poly-L-lysine is known to inhibit FAs maturation[33] and indeed, we could confirm that HeLa cells plated on such a substrate displayed fewer and smaller FAs (Supplementary Fig. 10a–c). Cells expressing TfR-mCherry and Erk-GFP were treated with control or β5-integrin-specific siRNAs and plated onto control or anti-mCherry antibody-coated glass. We observed that Erk-GFP nuclear enrichment, which reflects Erk activation, was similar for control cells plated on irrelevant or on anti-mCherry-coated surfaces (Fig. 4e, f). Confirming our western-blot

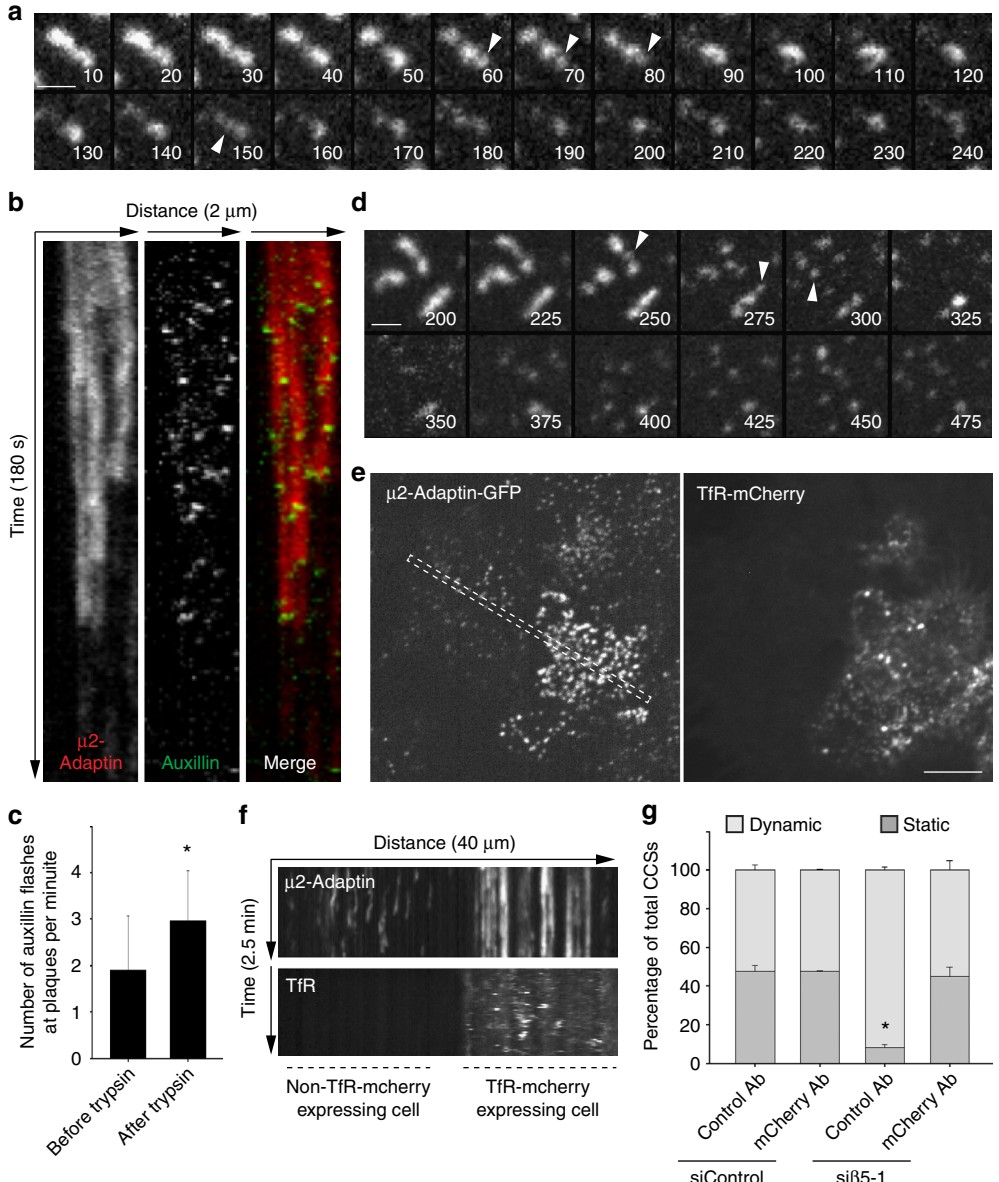

**Fig. 3** Clathrin-coated plaques assemble as a consequence of frustrated endocytosis. **a** Genome-edited HeLa cells expressing endogenous mCherry-tagged μ2-adaptin, seeded on collagen-coated glass, were treated with trypsin and imaged by spinning disk microscopy every 5 s. Time after trypsin addition is indicated in seconds. Arrowheads point to dot-like structure emanating from the disassembling plaques. Scale bar: 1 μm. **b** Kymographs showing plaque disassembly dynamics and concomitant GFP-auxillin bursts in HeLa cell treated and imaged as in **a**. Note that loss of μ2-adaptin-associated fluorescence is correlated with auxillin flashes. **c** Quantification of the number of auxillin flashes per plaque and per minute in HeLa cells before and after incubation with trypsin. Results are expressed as mean ± SD (*P < 0.001, Mann–Whitney rank sum test. A total of 90 structures from three independent experiments was quantified). **d** Genome-edited HeLa cells expressing endogenous mCherry-tagged μ2-adaptin, seeded on collagen-coated glass, were treated with Cilengitide and imaged by spinning disk microscopy every 5 s. Time after Cilengitide addition is indicated in seconds. Arrowheads point to dot-like structure emanating from the disassembling plaques. Scale bar: 1 μm. **e** Genome-edited HeLa cells treated with β5-specific siRNA and transfected with a construct encoding mCherry-tagged TfR were seeded on anti-mCherry antibodies-coated glass and imaged 24 h later by spinning disk microscopy every 5 s. Scale bar: 10 μm. **f** Kymograph showing CCS dynamics in the region corresponding to the boxed area in **e**. Note that the cell on the left is not transfected by the TfR-mCherry construct and only display dynamic CCSs. **g** Quantification of the dynamics of CCSs observed as in **e** in genome-edited HeLa cells expressing the TfR-mCherry construct and treated as indicated (*P < 0.001, one-way analysis of variance—ANOVA. n = 3). Results are expressed as mean ± SD

analysis, β5-depleted cells showed a reduction of Erk accumulation in the nucleus when plated on the control substrate (Fig. 4e, f). However, nuclear Erk accumulation was restored when plaque formation was allowed by plating cells on anti-mCherry antibody-coated glass (Fig. 4e, f). This results suggest that clathrin-coated plaques regulate steady-state Erk activity independently of β5-integrin. Accordingly, we observed that CHC and AP-2 were required to restore Erk nuclear levels in these conditions (Fig. 4f).

Together, our data demonstrate that plaques serve as signaling platforms.

We next aimed at investigating the mechanisms of plaque-regulated Erk activation. Confirming previous findings[34,35], we observed that the epidermal growth factor receptor (EGFR) accumulates at plaques upon stimulation (Supplementary Fig. 10d) and the hepatocyte growth factor receptor (HGFR) behaved similarly (Supplementary Fig. 10e). Large AP-2-positive

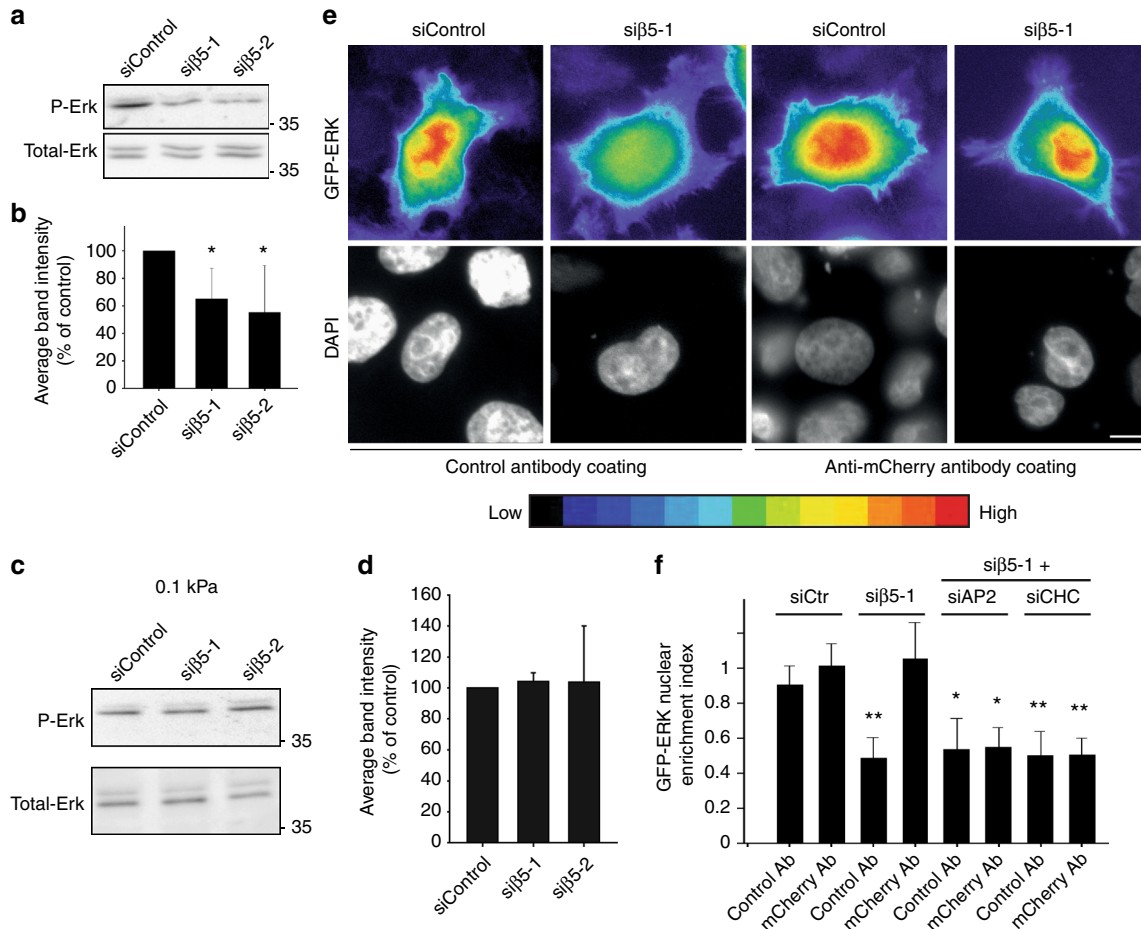

**Fig. 4** Clathrin-coated plaques regulate stiffness-dependent Erk signaling. **a** Western-blot analysis of phospho-Erk (P-Erk) levels in HeLa cells growing on collagen-coated glass and treated with control or β5-specific siRNAs as indicated (representative image of four independent experiments). Total-Erk was used as a loading control. **b** Densitometry analysis of bands obtained in western-blots as in **a**. Results are expressed as mean ± SD from four independent experiments (*P < 0.05, one-way analysis of variance—ANOVA). **c** Western-blot analysis of phospho-Erk (P-Erk) levels in HeLa cells growing on collagen-coated, 0.1 kPa polyacrylamide gels and treated with control or integrin β5-specific siRNAs, as indicated. Total-Erk was used as a loading control (representative image of three independent experiments). **d** Densitometry analysis of bands obtained in western-blots as in **c**. Results are expressed as mean ± SD from three independent experiments. **e** GFP-Erk-expressing HeLa cells treated or not with a β5-specific siRNA and transfected or not with TfR-mCherry, as indicated, were seeded on anti-mCherry antibodies-coated glass and imaged 24 h later. Scale bar: 10 μm. A color-coded scale for low and high signal intensity is shown. **f** Quantification of GFP-Erk nuclear enrichment index in cells as in **e** and cultured on control antibody- or anti-mCherry antibody-coated glass, as indicated, and treated or not with indicated siRNAs. Results are expressed as mean ± SD (*P < 0.01, **P < 0.001, ANOVA. ControlAb-siCtr: 114 cells from n = 5 independent experiments; mChAb-SiCtrl: 119 cells from n = 5 independent experiments; ControAb-siβ5-1: 146 cells from n = 5 independent experiments; mChAb-Siβ5-1: 120 cells from n = 5 independent experiments; ControlAb-siβ5-1 + siAP-2: 69 cells from n = 3 independent experiments; mChAb-siβ5-1 + siAP-2: 68 cells from n = 3 independent experiments; ControlAb-siβ5-1 + siCHC: 63 cells from n = 3 independent experiments; mChAb-siβ5-1 + siCHC: 67 cells from n = 3 independent experiments)

structures corresponding to plaques were strongly marked with an anti-phosphotyrosine antibody upon EGF stimulation (Fig. 5a, b). Smaller structures corresponding to CCPs were also positive for the anti-phosphotyrosine staining but to a lesser extend (Fig. 5a, b). Anti-phosphotyrosine staining was absent from both large and small CCSs when cells were first treated with Gefitinib, a drug that inhibits EGFR kinase activity (Fig. 5a, b). Thus, our data suggest that plaques may serve as signaling platforms in the EGFR pathway. Accordingly, EGF-induced Erk activation was reduced upon plaques ablation in β5-integrin-depleted cells (Fig. 5c, d). Of note, cell-surface EGFR levels where not significantly modulated in these later conditions, excluding the possibility that reduced Erk activity is due to a down-modulation of the surface-exposed receptor (Supplementary Fig. 11a). In addition, a specific enrichment of phosphotyrosine staining at plaques was also observed upon stimulation with HGF

(Supplementary Fig. 11b, c). Collectively, these data strongly suggest that plaques serve as signaling platforms for different receptors. In addition to EGFR and HGFR, many other signaling receptors are known to be recruited at plaques[11,12], and we propose that this is critical to regulate the observed steady-state Erk activity. Indeed, we observed that EGFR and HGFR are still recruited to immobilized TfR-induced plaques in the absence of β5-integrin and that phosphotyrosine staining still accumulate at these artificial structures upon stimulation (Fig. 5e–g). Thus, our model is that αvβ5 controls plaque formation on stiff environments but that it is the subsequent recruitment of different signaling receptors at plaques that is the cause of Erk activity modulation.

**Plaques regulate cell proliferation.** Our observations suggest that plaques are signaling platforms that assemble in response to

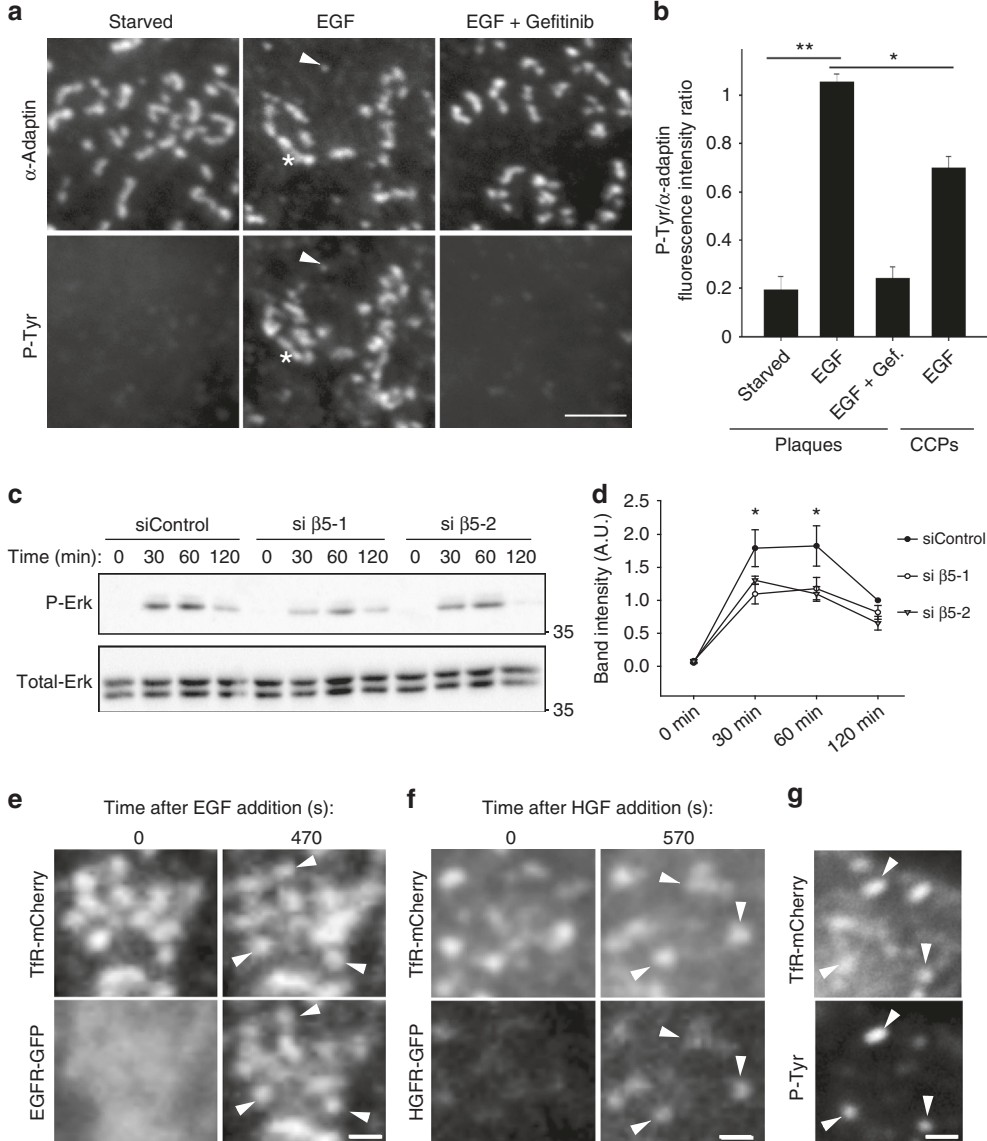

**Fig. 5** Clathrin-coated plaques locally regulate receptor-dependent Erk signaling. **a** HeLa cells seeded on collagen-coated glass were starved for 4 h and then treated or not with 10 ng/ml EGF for 5 min alone or added after 30 min preincubation with 10 μM Gefitinib. Cells were then fixed and stained for α-adaptin and phosphotyrosines. The arrowhead points to one CCP and the star marks a plaque. Scale bar: 2 μm. **b** Quantification of phosphotyrosines accumulation at plaques or CCPs in the indicated conditions. Results are expressed as mean ± SD (*$P < 0.05$, **$P < 0.001$, one-way analysis of variance—ANOVA. $n = 3$. Number of structures analyzed per condition: Starved 160 plaques; EGF: 452 plaques, EGF + Gefitinib: 301 plaques, CCPs/EGF: 200 pits). **c** Western-blot analysis of phospho-Erk (P-Erk) levels in starved HeLa cells treated with control or β5-specific siRNAs as indicated, and stimulated for the indicated time with 10 ng/ml EGF. Total-Erk was used as a loading control (representative image of three independent experiments). **d** Densitometry analysis of bands obtained in western-blots as in **c**. Results are expressed as mean ± SD (*$P < 0.05$, ANOVA. $n = 3$). **e** HeLa cells treated with β5-specific siRNAs were transfected with plasmids encoding for TfR-mCherry and EGFR-GFP and seeded on anti-mCherry antibodies-coated glass. Cells were serum-starved for 4 h and then treated with 10 ng/ml EGF for 470 s. Scale bar: 0.5 μm. Arrowheads point to plaques positive for EGFR-GFP. **f** HeLa cells treated with β5-specific siRNA were transfected with plasmids encoding for TfR-mCherry and HGFR-GFP and seeded on anti-mCherry antibodies-coated glass. Cells were serum-starved for 4 h and then treated with 50 ng/ml HGF for 570 s. Scale bar: 0.5 μm. Arrowheads point to plaques positive for HGFR-GFP. **g** HeLa cells treated with β5-specific siRNA were transfected with plasmids encoding for TfR-mCherry and seeded on anti-mCherry antibodies-coated glass. Cells were serum-starved for 4 h and then treated with 10 ng/ml EGF for 5 min prior to fixation and staining for phosphotyrosines. Scale bar: 1.5 μm. Arrowheads point to plaques positive for P-Tyr

substrate rigidity. Because their formation does not require a functional acto-myosin machinery (Fig. 1), plaques may represent a distinct, contractility-independent mechanotransduction system. Along this line, blebbistatin or cytochalasin treatments did not prevent phosphotyrosine accumulation at plaques upon EGF stimulation (Fig. 6a, b). Erk activity has been reported to be mechanosensitive in a manner that is independent of acto-myosin

forces[36,37]. Indeed, we observed that phosphorylated-Erk levels were reduced on soft as compared to hard environment (Fig. 6c, d) and inhibiting myosin-dependent contractility using blebbistatin did not modulate Erk activation status (Fig. 6e, f). Thus, Erk activation on rigid environments depends on the presence of clathrin-coated plaques (Fig. 4) but not on cell contractility. Accordingly, Erk activity was increased when cells were seeded on

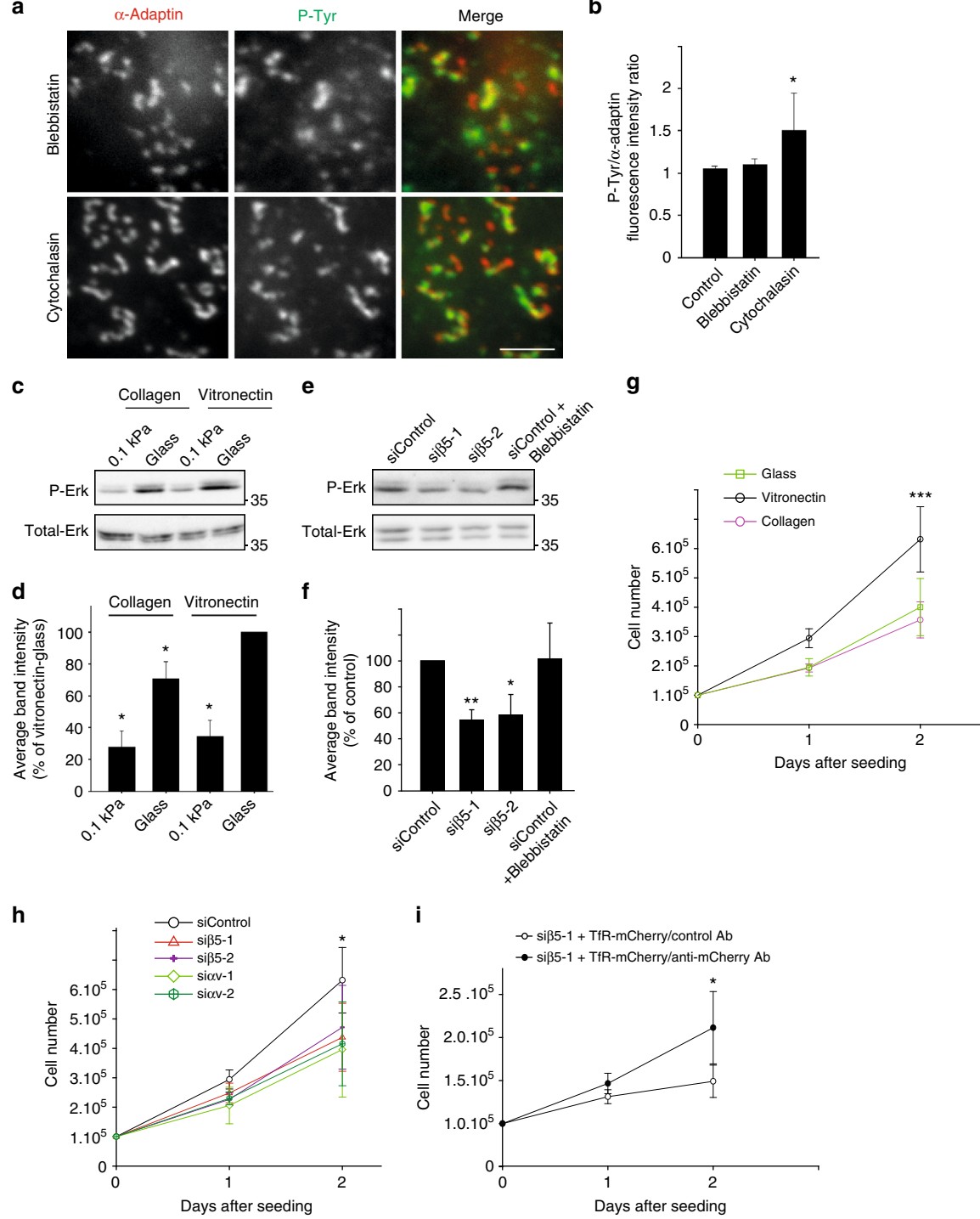

vitronectin-coated glass, a condition that favors the formation of plaques (Fig. 6c, d). However, vitronectin coating does not overcome the need for a stiff environment in order for plaques to assemble (Supplementary Fig. 7a–d) and thus, it did not modulate Erk activation status on 0.1 kPa gels (Fig. 6c, d). In agreement with the role of Erk in controlling cell proliferation, we noticed that cells proliferated faster on vitronectin- as compared to collagen-coated glass (Fig. 6g) and inhibiting plaque formation with β5-specific or αv-specific siRNAs reduced proliferation rate (Fig. 6h). In addition, rescuing plaque formation in β5-integrin-depleted cells through immobilizing the TfR on the substrate resulted in an increased proliferation rate (Fig. 6i). Thus, plaque

assembly on rigid environments results in increased Erk activity and sustained cell proliferation.

## Discussion

Overall, we showed that clathrin-coated plaques are mechanosensitive signaling platforms that assemble on rigid substrates as a consequence of frustrated endocytosis. Clathrin-coated plaques have been consistently observed in diverse cell types but have overall received little attention as compared to canonical CCPs, possibly reflecting the lack of specific markers for these structures. Yet, past investigations have pointed to a possible role of plaques in cell adhesion because of the close association of these

**Fig. 6** Signaling at plaques is contractility-independent and regulate cell proliferation. **a** HeLa cells on collagen-coated glass were treated for 1 h with 10 μM Blebbistatin or 10 μM Cytochalasin D prior to be fixed and stained for α-adaptin (red) and phosphotyrosines (P-Tyr, green). Scale bar: 3 μm. **b**, Quantification of phosphotyrosines accumulation at plaques in cells treated as in a, as indicated (*$P < 0.05$, one-way analysis of variance—ANOVA. Control: 402 plaques from $n = 3$ independent experiments; Blebbistatin: 303 plaques from $n = 3$ independent experiments; Cytochalasin: 302 plaques from $n = 3$ independent experiments). **c** Western-blot analysis of phospho-Erk (P-Erk) levels in HeLa cells cultured on collagen- or vitronectin-coated glass or 0.1 kPa polyacrylamide gels, as indicated. Total-Erk was used as a loading control (representative image of three independent experiments). **d** Densitometry analysis of bands obtained in western-blots as in **c** (*$P < 0.05$, ANOVA. $n = 3$). **e** Western-blot analysis of phospho-Erk (P-Erk) levels in HeLa cells growing on collagen-coated glass and treated with control or integrin β5-specific siRNAs and incubated or not with 10 μM Blebbistatin for 1 h, as indicated. Total-Erk was used as a loading control (representative image of three independent experiments). **f** Densitometry analysis of bands obtained in western-blots as in **e** (*$P < 0.05$, **$P < 0.01$, ANOVA. $n = 5$). **g** Equal numbers of HeLa cells were plated on non-coated glass (open squares), or on collagen-coated (purple open circles) or vitronectin-coated glass (black open circles), as indicated. Cells were harvested and counted 24 and 48 h after plating (***$P < 0.001$, ANOVA. $n = 3$). **h** HeLa cells treated with control (circles), β5-specific (triangles and crosses), or αv-specific siRNAs (diamonds and hexagons) for 48 h were seeded on vitronectin-coated glass in equal numbers. 24 and 48 h later, cells were harvested and counted (*$P < 0.05$, ANOVA. $n = 3$). **i** HeLa cells treated with β5-specific siRNA were transfected with a plasmid encoding TfR-mCherry and seeded in equal numbers on glass coated with either anti-mCherry (black circles) or control antibodies (open circles). 24 and 48 h later, cells were harvested and counted (*$P < 0.05$, ANOVA. $n = 3$). All results are expressed as mean ± SD

structures with the substrate[16], and because they were shown to be enriched in integrins, and in particular in β5-integrin[10]. We report here that αvβ5-integrin is required for plaque formation and can be considered as a marker of these structures although it can also be found at FAs in some cases. αvβ5 engagement with the substrate prevents the clathrin machinery to bud. This is reminiscent of the β1-integrin-mediated frustrated endocytosis of clathrin-coated tubes found on collagen fibers[9]. However, in this later case, frustration was only transient while plaques can be extremely long-lived. It has recently been reported that actin polymerization can participate in plaque disassembly upon LPA-receptor activation[12]. It is possible that actin polymerization provides an extra force needed for the clathrin machinery to overcome αvβ5-mediated adhesion and to bud at plaques. Along this line, inhibiting actin polymerization resulted in a slight increase in plaque size (Fig. 1e). Together, these data suggest that a force balance between endocytosis and adhesion regulates the dynamics of these structures. The equilibrium between these forces is modulated by the rigidity of the substratum so that plaques can only assemble on relatively stiff substrates.

Plaques have also been proposed to have a role in signal transduction[11,12] although direct evidences have been missing because of the lack of tools to specifically impair plaque assembly. How plaques regulate signaling is not clear yet. Others have proposed that CCSs are platforms for signaling[31], independently of endocytosis. The capacity of CCSs to compartmentalize the plasma membrane is certainly an important, underestimated factor that will need to be investigated in the future. In that respect, large CCSs such as plaques may be more efficient in transducing the signal.

Because the formation of these structures does not rely on a functional acto-myosin network, we propose that plaques represent an alternative, contractility-independent rigidity sensing mechanism. The use of different strategies to sense the environment's elasticity may reflect the importance of this mechanical parameter that has fundamental biological consequences, ranging from cell proliferation to differentiation. Because αvβ5-integrin and vitronectin levels, together with tissue rigidity, are often modulated in different physio-pathological conditions[38–42], CCSs-dependent mechanotransduction may have a universal role in regulating the cell response to changing environmental conditions.

## Methods

**Cell lines and constructs.** HeLa cells (a gift from P. Chavrier, Institut Curie, Paris, France; ATCC CCL-2), genome-edited HeLa cells engineered to expressed an endogenous GFP-tagged or mCherry-tagged μ2 subunit, HepG2 cells (ATCC HB-8065), Caco-2 cells (ATCC HTB-37), MDA-MB-231 cells (a gift from P. Chavrier, Institut Curie, Paris, France; ATCC HTB-26), or genome-edited MDA-MB-231

cells engineered to expressed an endogenous GFP-tagged μ2 subunit (a gift from D. Drubin, University of California-Berkeley, California, USA) were grown in DMEM Glutamax supplemented with 10% foetal calf serum at 37 °C in 5% CO$_2$. All cell lines have been tested for mycoplasma contaminations. For most experiments, cells were grown on substrates coated with a 50 μg/ml solution of collagen-I (Thermo Fisher Scientific—Cat. Nr. A10483-01) unless otherwise stated. DNA sequence encoding full-length β5-Integrin was obtained by PCR by using the cDNA of human β5-Integrin, a gift from Raymond Birge (Addgene plasmid #14996) as a template. PCR fragments with engineered flanking restriction sites (XhoI/BamHI) were subcloned into the multi-cloning sites of pEGFP-N1 (Clontech) to encode an in-frame fusion protein with the carboxy-terminal EGFP-tag (pEGFP-N1- Integrin β5). SiRNA-resistant Integrin β5 was obtained by site-directed mutagenesis of the β5-encoding cDNA at the following positions: A69T, T72G, C73G, C75G (silent mutations—resistant to siβ5-1). mCherry-TfR was a gift from Michael Davidson (Addgene plasmid #55144). GFP-Erk2 was a gift from Dr.Hesso Farhan. EGFR-GFP was a gift from Alexander Sorkin (Addgene plasmid # 32751). pLenti-MetGFP was a gift from David Rimm (Addgene plasmid #37560). mEmerald-Alpha-V-Integrin was a gift from Michael Davidson (Addgene plasmid #53985).

Plasmids were transfected 24 h after cell plating using either Lipofectamine 3000 according to the manufacturer's instructions or electroporating cells in suspension using AMAXA nucleofector Kit V according to the manufacturer's instructions. Alternatively, linear PEI (MW 25.000—Polysciences Cat. Nr. 23966) at 1 mg/ml was used to transfect 50% confluent cells in a 6 well plate according to the following protocol: 2 μg of DNA were added to 100 μl of OptiMEM, followed by addition of 4 μl of PEI, vortex and incubation for 10 minutes at RT prior to add the mix to the cells.

**Antibodies and drugs.** Mouse monoclonal anti-clathrin heavy chain (CHC – Cat. Nr. 610500) antibody and mouse monoclonal anti-FAK (Cat. Nr. 610088) antibody were obtained from BD Transduction Laboratories (Becton Dickinson France SAS, Le Pont-De-Claix, France). Rabbit polyclonal anti-actin was from Sigma (Cat. Nr. A5060). Rabbit polyclonal anti-α-adaptin antibodies (M300), rabbit polyclonal DAB2 (Cat. Nr. sc-13982), Goat polyclonal anti-Talin1 (C20) were purchased from Santa Cruz Biotechnology Inc. (Santa Cruz, CA, USA). Integrin αvβ5 (Cat. Nr. MAB1961) and P-Tyrosine (Cat. Nr. 05-321) were obtained from Millipore. Mouse monoclonal anti-α-adaptin (Cat. Nr. ab2807) was purchased from Abcam. Numb (Cat.Nr. 2756), Integrin β5 (Cat. Nr. 3629), Tot-ERK1/2 (Cat. Nr. 9102) and P-ERK1/2 (Cat. Nr. 9101) were purchased from Cell Signalling. Tot-ERK1/2 (Cat. Nr. 13-6200) was purchased from Thermo Fisher. Antibodies used for western-blot analyses were diluted at 1:1000 in PBS-0.1% Tween-5% BSA or 5% non-fat dried milk. For immunofluorescences, antibodies were diluted 1:200 in PBS-0.3% BSA. HRP-conjugated anti-mouse and anti-rabbit antibodies for western-blot were from Jackson ImmunoResearch Laboratories (West Grove, PA, USA) and were used at a dilution of 1:3000. Alexa-conjugated antibodies as well as Cy3 and Cy5-conjugated antibodies were from Molecular Probes (Invitrogen) and were used at a dilution of 1:200. For expansion microscopy, the following secondary antibodies were used: Donkey anti-Mouse IgG (H + L) Highly Cross-Adsorbed Secondary Antibody, Alexa Fluor® 488, Thermo Fisher (Cat. Nr. A21202), Goat Anti-Rabbit IgG H&L, Alexa Fluor® 568, AbCam (Cat. Nr. ab175471), CF®633 Donkey Anti-Mouse IgG (H + L), highly cross-adsorbed, Biotium (Cat. Nr. 20124) and were used at a dilution of 1:100. The anti-mCherry and P-FAK Tyr576 (western-blot) antibodies were obtained from the recombinant antibody platform of the Curie Institute, Paris and was used at a dilution of 1:1000. P-FAK Tyr397 (IF) was a gift from Monique Arpin and was used at a dilution of 1:200. Alexa Fluor® 488 EGF complex was obtained from Thermo Fisher (Cat. Nr. E-13345). Rat tail Collagen-I (Cat. Nr. A10483-01) and Vitronectin (Cat. Nr. A14700) were purchased from GIBCO. Human recombinant EGF (Cat. Nr. E9644) and HGF (Cat. Nr. 1404) were purchased from Sigma. Blebbistatin (Cat. Nr. B0560), Gefitinib (Cat. Nr. CDS022106)

and Cytochalasin D (Cat. Nr. 8273) were purchased from Sigma. Latrunculin A (Cat. Nr. ab144290) was purchased from AbCam. Blebbistatin, Cytochalasin D, and Gefitinib were used at a final concentration of 10 μM and Latrunculin A at 5 μM. Cilengitide was purchased from Selleckchem (Cat. Nr. S7077) and used at a final concentration of 10 μM. For western-blot experiments, cells were lysed in ice-cold MAPK buffer (100 mM NaCl, 10 nM EDTA, 1% IGEPAL® CA-630, 0.1% SDS, 50 mM TRIS–HCl pH 7.4) supplemented with protease and phosphatase inhibitors.

**RNA interference**. For siRNA depletion, 200,000 cells were plated in six well plates. After 24 h, cells were treated with the indicated siRNA (30 nM) using RNAimax (Invitrogen, Carlsbad, CA) according to the manufacturer's instruction. Protein depletion was maximal after 72 h of siRNA treatment as shown by immunoblotting analysis with specific antibodies. To deplete CHC or μ2-adaptin, cells were transfected once as described above and then a second time, 48 h later, with the same siRNAs. In this case, cells were analyzed 96 h after the first transfection. The following siRNAs were used: β5-1, 5′-GCUCGCAGGUCUCAACA UA-3′; β5-2, 5′-GGUCUAAAGUGGAGUUGUC-3′; μ2-adaptin, 5′-AAGUGGA UGCCUUUCGGGUCA-3′; Clathrin heavy chain (CHC), 5′-GCUGGGAA AACUCUUCAGATT-3′; αv-1, 5′-CCUCUGACAUUGAUUGUUA-3′; αv-2, 5′-C CGAAACAAUGAAGCCUUA-3′; DAB2, 5′-GAGCAUGAAACAUCCAGAU AATT-3′; Numb, 5′-GAUAGUCGUUGGUUCAUCATT-3′; Integrin β1 ON-TARGET plus SMART pool (Dharmacon L-004506-00), Integrin β3 siGENOME Human ITGB3 siRNA (Dharmacon M-004124-02); Talin1, 5′-AC AAGAUGGAUGAAUCAAUUUU-3′; non-targeting siRNAs (siControl), ON-TARGET plus Non-Targeting SMART pool siRNAs (Dharmacon D-001810-01).

**Reverse transcription and polymerase chain reaction**. Total RNA was isolated from HeLa cells by using the Qiagen RNeasy kit. cDNA was prepared from 100 ng total RNA by using the high capacity cDNA reverse transcription kit (Applied Biosystems) according to the manufacturer's instructions. Q-PCR was performed by using Fast SYBR green PCR MasterMix (Applied Biosystems). Reverse transcription and polymerase chain reaction (RT-PCR) were run on a QuantStudio™ 7 Flex Real-Time PCR system (Applied Biosystems). Expression of each gene was normalized to the expression of GAPDH (primers: Fw: 5′-CTTTTGCGTCGC CAGCCGAG-3′; Rev 5′- CCAGGCGCCCAATACGACCA-3′). Specific primers for Integrin β5 were designed using the free online tool PrimerExpress® (Fw: 5′-CT GGAACAACGGTGGAGATT-3′; Rev: 5′-TACCCCATCTTGGCAGGTAG-3′). The relative amount of integrin β5 cDNA was normalized to the cDNA relative amount of the housekeeping gene GAPDH. Values obtained in the control were set to 1 and the other values expressed as percentage of control. The reaction was performed three times and the data are expressed as mean ± SD.

**Immunofluorescence microscopy and fluorescence quantification**. Cells were fixed in ice-cold methanol unless stated otherwise and processed for immuno-fluorescence microscopy by using the indicated antibodies. For anti-Phosphotyrosine staining (P-Tyr), cells were briefly extracted for 30 s using 0.1% Triton prior to fixation. Depending on experiments imaged either through a ×100 1.40NA UPlanSApo objective lens of a wide-field IX73 microscope (Olympus) equipped with an Orca-Flash2.8 CMOS camera (Hamamatsu) and steered by CellSens Dimension software (Olympus), or by spinning microscopy (see description below). Surface levels of EGFR were measured by incubating the cells on ice with A488-EGF for 1 h. Cells were subsequently fixed with PFA 4% and images were taken. Total fluorescence was measured on at least 50 cells per condition in three independent experiments. Data are expressed as mean ± SD.

For expansion microscopy[43], cells grown on coverslips were immunostained and then treated for 10 min with glutaraldehyde 0.25%. The coverslips were washed and incubated in monomer solution for 1 min (1 × PBS, 2 M NaCl, 2.5% (w/w) acrylamide, 0.15% (w/w) N,N′-methylenebisacrylamide, 8.625% (w/w) sodium acrylate). 120 μl drops of Gel solution (1.3 μl of TEMED, 1.3 μl of APS 10%, 197.4 μl of monomer solution) were deposited on parafilm and the coverslips were deposited on top. After 30 min, gels were moved to the digestion solution (Tris-acetate-EDTA (TEA), 0.5% Triton-×100, 0.8 M guanidine HCl, Proteinase K—8 U/ml added fresh before use) for 30 more minutes. Gel were then manually separated from the coverslips and incubated in ddH₂O for at least 2 h, until full expansion (4.167 expansion factor). All quantifications on immunofluorescence images were done with FIJI after background subtraction. To measure the size of CCSs on expansion microscopy images, α-adaptin-positive structures were manually delimited and the area was measured using ImageJ. Dot-like objects with area lower or equal to 0.0175 μm² (radius 0.075 μm) were classified as clathrin-coated pits (CCPs), whereas irregularly shaped objects with an area larger than 0.0175 μm² were classified as clathrin-coated plaques. The area of the adherent side of the cell was also measured in order to calculate the percentage of adherent surface occupied by plaques. Data are expressed as mean ± SD.

To analyze focal adhesions (FAs), FAK-positive structures were segmented in ImageJ excluding objects with radius smaller than 75 nm. At least 160 FAs were counted for each condition from nine cells per experiments from three independent experiments. Data are expressed as mean ± SD. Fluorescence enrichment index of αvβ5 or of P-Tyr at CCPs or plaques was measured by manually delimiting α-adaptin-positive structures in order to measure the

fluorescence intensity of α-adaptin and of αvβ5 or of P-Tyr staining. Values were subsequently normalized to the average fluorescence of the α-adaptin channel and normalized intensity values were used for the analysis. At least 100 structures (50 for the HGF stimulated cells) per condition and per experiments were counted in three independent experiments. Data are expressed as mean ± SD. For the data in Supplementary Figure 9, phosphotyrosine (P-Tyr)-positive structures were segmented in ImageJ and mean fluorescence intensity was measured in at least 15 cells per condition in three independent experiments. Data are expressed as mean ± SD. For measuring ERK-GFP intensity in the nucleus, cells cytoplasm and nuclei were manually segmented, GFP-associated integrated intensity was measured in the cytoplasm and in the nucleus and the nucleus/cytosol ratio was calculated for every cell. At least 15 cells per condition were measured in at least three independent experiments. Results are expressed as the relative enrichment over the no-enrichment condition (nucleus/cytosol signal = 1) ± SD.

**Antibody coating**. Coverslips were incubated in a sterile solution of sodium bicarbonate (0.1 M, pH 9.5) for 1 h at 37 °C. Coverslips were then incubated with poly-L-lysine 0.01% (Sigma, diluted in water) for 1 h at 37 °C, washed once with PBS, dried and incubated overnight at 37 °C with the desired antibody (mCherry antibody or anti-rabbit A647 from Invitrogen; final concentration of 1.25 μg/ml) diluted in the bicarbonate solution. Acrylamide gels were first coated with Protein G (Sigma) diluted in sodium bicarbonate (0.1 M, pH 9.5) to a final concentration of 5 μg/ml for 1 h at 37 °C, then treated with poly-L-lysine 0.01% (Sigma, diluted in water) for 1 h at 37 °C, washed once with PBS, dried and incubated overnight at 37 °C with the desired antibody.

**Acrylamide gels of controlled stiffness**. Coverslips or fluorodishes were incubated with APTMS (3-aminopropyltrimethoxysilane) for 15 min at RT, then washed extensively with water and incubated for 30 min with Glutaraldehyde 0.5% in PBS and washed again with water. Acrylamide 40% and bis-acrylamide 2% were mixed (respectively 5% and 0.04% for 0.1 kPa gels, 7.5% and 0.06% for 5 kPa gels, 18% and 0.4% for 31.7 kPa gels, and 16% and 0.96% for 80 kPa) with PBS, APS 10% and TEMED. 9 μl of this solution were polymerized on the treated glass. Gels were washed with PBS, followed by a 30 min incubation with 300 μl 50 mM Hepes pH 7.5 + 100 μl sulfo-sampah (1 mg/ml in 50 mM Hepes pH 7.5) + 100 μl EDC (10 mg/ml in 50 mM Hepes pH 7.5). Gels were subsequently cross-linked under UV light for 10 min, washed and incubated with 50 μg/ml collagen-I for 1 h at 37 °C. Elasticity of the different gels was controlled by Atomic Force Microscopy as indicated in Betz et al.[44] The generation of 80 kPa gels was performed according to a previously published protocol[45].

**Micropatterns**. Coverslips were cleaned by washing in 70% ethanol and subsequent irradiated under UV light for 5 min. The activated side of the coverslips was then covered with 0.1 mg/ml PLL-g-PEG (Surface Solutions, Zurich) for 1 h at RT. Coverslips were then washed twice in water and ring-shaped areas were exposed to deep UV during 5 min using a photomask. Coverslips were recovered and coated with Alexa-568-labeled Collagen-I at 50 μg/ml overnight at 37 °C. The following day, 50-80,000 HeLa cells were plated on micropatterns in complete medium supplemented with penicillin and streptomycin. Cells were either fixed or imaged 5–6 h after plating. Eight to ten cells per experiment were measured per experiments in three independent experiments. Data are expressed as mean ± SD.

**Electron microscopy of unroofed cells**. HeLa cells were plated on Poly-L-lysine coated coverslips in 12 well plates. 24 h after plating, medium was aspirated and replaced with 4 ml of stabilization buffer (3 mM EGTA, 5 mM MgCl₂, 70 mM KCl, 30 mM HEPES pH 7.4). To unroof cells, 1 ml of 2% PFA was pipetted directly on the cells with a pipette kept perpendicular to the glass at 1 mm from the surface. The coverslips were then immediately transferred to 2% glutaraldehyde for 30 min and then left in Cacodylate buffer.

Scanning electron microscopy observations of unroofed HeLa cells were performed as previously described[46]. Briefly, fixed HeLa cells were dehydrated in a series of increasing ethanol concentrations. Critical point was dried using carbon dioxide in a Leica EMCPD300. After coating with 2 nm platinum, cells were examined with a FEI Quanta FEG250 scanning electron microscope.

**Spinning disk microscopy of live cells**. Cells were imaged at 5 s intervals for the indicated time using a spinning disk microscope (Andor) based on a CSU-W1 Yokogawa head mounted on the lateral port of an inverted IX-83 Olympus microscope equipped with a ×60 1.35NA UPLSAPO objective lens and a laser combiner system, which included 491 and 561 nm 100 mW DPSS lasers (Andor). Images were acquired with a Zyla sCMOS camera (Andor). The system was steered by IQ3 software (Andor).

For CCS dynamics quantification, we measured the lifetime of CCSs using the TrackMate plugin of ImageJ[47]. Tracks below 5 s of duration (detected on only 1 frame) were discarded. Measured individual lifetimes were pooled into two groups: the "dynamic" group corresponding to structures with a lifetime below the duration of the movie (5 min) and the "static" group with a lifetime of 5 min. Of note, the relative percentage of dynamic versus static structures depends on the duration of the movie because static structures are only counted once while

dynamic structures continuously nucleate and disappear during the movie. For this reason, all quantifications of CCS dynamics represent the relative number of static or dynamic events detectable at the plasma membrane at a given time point. At least 1000 CCSs from at least five cells per conditions and per experiments were tracked in 3–5 independent experiments. Data are expressed as mean ± SD.

To measure Auxillin bursts, cells were imaged at 1 frame every 2 s for 3 min. Then, trypsin to a final concentration of 0.0375% was added and imaging was immediately started until disappearance of plaques. Ninety structures (plaques) from a total of six cells were analyzed from two independent experiments. Data are expressed as mean ± SD.

**Total internal reflection fluorescence microscopy**. For total internal reflection fluorescence microscopy (TIRF), HeLa cells transfected with the indicated plasmids were imaged through a ×100 1.49 NA TIRF objective lens on a Nikon TE2000 (Nikon France SAS, Champigny sur Marne, France) inverted microscope equipped with a QuantEM EMCCD camera (Roper Scientific SAS, Evry, France/Photometrics, AZ, USA), a dual output laser launch, which included 491 and 561 nm 50 mW DPSS lasers (Roper Scientific), and driven by Metamorph 7 software (MDS Analytical Technologies, Sunnyvale, CA, USA). A motorized device driven by Metamorph allowed the accurate positioning of the illumination light for evanescent wave excitation.

**Stimulated-emission-depletion microscopy**. Image acquisitions were performed with a ×100 oil immersion objective (NA 1.4) through gated Continuous Wave (gCW) Stimulated-emission-depletion (STED) imaging (TCS SP8-3×; Leica Microsystems) with optimized parameters for Alexa Fluor 568 detection. Samples (zoom 4, pixel size = 14 nm) were excited with a 575 nm wavelength of a supercontinuum laser and a 660-nm laser for depletion. For Alexa Fluor 568, 30% AOTF, conventional scanner (400 Hz, Line Average 2, Accumulation 4) and 50% of depletion lasers were used. Fluorescence (585–630 nm) was collected with a hybrid detector (Gain 30%) in the gated mode (0.5–6 ns) and a pinhole for 1 Airy Unit. Deconvolution of raw data from STED imaging was obtained through image processing with Huygens professional 4.5.1 software.

**Cell proliferation assays**. Cells were seeded at a density of 100,000 per well in 12 wells plates coated with 50 μg/ml collagen-I, or 0.5 μg/ml vitronectin, or uncoated. Twenty-four and forty-eight hours later, cells were harvested and counted. All conditions were plated in duplicates in at least three independent experiments. Data are expressed as mean ± SD.

**Statistical analyses**. Statistical analyses in Figs. 1, 2c, e, 3g, 4d, f, 6b, f, g, h, i, Supplementary Figs. 2, 4, 5d, 6b, c, e, 7d, g, 8, 9d, 11a have been performed using one-way analysis of variance (ANOVA) followed by all pairwise multiple comparison procedure (Holm–Sidak method). Statistical analyses in Figs. 4b, 5b, d, 6d, h and Supplementary Figs. 6g, 7b, 9f have been performed using ANOVA followed by all pairwise multiple comparison procedure (Neuman–Keuls method). Mann–Whitney rank sum test was used in Fig. 3c and Supplementary Data Fig. 5c. Student's t-test was used in Fig. 2b and Supplementary Figs. 10 and 11c. All data are presented as mean of at least three independent experiments ± SD. All statistical analyses were performed using SigmaPlot software.

## Data availability
The authors declare that all data supporting the findings of this study are available within the article and its supplementary information files or from the corresponding author upon reasonable request.

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

## Acknowledgements

We wish to thank Drs J. Ivaska, P. Chavrier, A. Benmerah, and C. Albiges-Rizo for critical comments on the manuscript. We thank the imaging facilities of Gustave Roussy, Institut Curie, and Institut Imagine for help with image acquisition. We thank Isabelle Fourqueaux (TRI imaging facility, CMEAB, Toulouse) for her help with sample preparation for scanning electron microscopy. Core funding for this work was provided by the Gustave Roussy Institute and the Inserm and additional support was provided by grants from ATIP/Avenir Program, la Fondation ARC pour la Recherche sur le cancer, Le Groupement des Entreprises Françaises dans la LUtte contre le Cancer (GEFLUC), and from the Agence Nationale de la Recherche (ANR-15-CE15-0005-03) to GM. F.B. was supported by a fellowship from La Ligue Nationale contre le Cancer.

## Author contributions

F.B. designed and performed experiments, analyzed results, and wrote the manuscript. S.D., N.E., and N.L. performed experiments. N.L. generated CRISPR knock-in cell lines. A.C. and K.S. designed experiments related to micropatterning technology. T.B. designed experiments and provided assistance in generating polyacrylamide gels. D.M.V. provided reagents and designed experiments. R.P. designed and performed electron microscopy analysis. G.M. supervised the study, designed experiments, and wrote the manuscript.
