## [Peer Review File · Nature Communications]

Reviewers' comments:

Reviewer #1 (Remarks to the Author):

The manuscript by Baschieri et al. demonstrates that flat clathrin-coated structures (CCSs) named plaques are contractility-independent mechanosensitive transduction hubs. In this regard, the Authors found that plaques generate upon substrate rigidity independent of actin and myosin-II activity. In particular, plaques originate from the elasticity-regulated affinity of $\alpha\beta5$ integrin for the ECM and serve as signaling platforms toward Erk activation and cell proliferation on stiff environments.

The matter is interesting and deserve further research roads and applications. However, I believe that it could be difficult for a general audience to follow the consecutive meaning of the experimental design as the Authors have written the manuscript. Moreover, many figures are shown in black and white, the readers would better appreciate the findings if shown in color.

Comments

The Authors ascertained that certain cells as HeLa, HepG2 and Caco2 cells display both flat clathrin structures called plaques and dynamic clathrin-coated pits, whereas MDA-MB-231 cells only the latter. The Authors no longer observed plaques in $\alpha\beta5$ depleted cells, hence it could be interesting to engineer MDA-MB-231 cells (and other cells displaying similar features as MDA-MB-231 cells) to express $\alpha\beta5$ in order to strengthen the results observed on the role of $\alpha\beta5$. In addition, if these results may be linked to the aggressive and metastatic behavior of MDA-MB-231 (and other cells), the data presented may be of particular interest toward the role of plaques in cancer progression. The Authors suggested an intriguing chance regarding plaques as promiscuous platforms for different growth factor-activated signaling, as plaques were strongly marked with an anti-phosphotyrosine antibody upon EGF and HGF stimulation. Thereafter, the Authors found that plaques regulate Erk signaling independently of $\beta5$ -integrin that is required for plaque formation and therefore EGFR-mediated signaling. The Authors could explain this result that appears contrasting with the classical transduction pathway linking EGFR with Erk signaling. In this regard, it could be interesting to evaluate upon EGF stimulation both Erk activation and the expression levels of c-fos, which is a well acknowledged molecular sensor of this signaling and closely related to cell proliferation.

Reviewer #2 (Remarks to the Author):

Cells internalize membrane proteins through the formation of clathrin-coated structures that can appear as dynamic clathrin-coated pits or long-lived structures called, clathrin lattices or plaques. These clathrin lattices have been suggested to be sites of adhesion and may act as signaling platforms, which concentrate activated signaling receptors including receptor tyrosine kinases (RTKs). The manuscript by Baschieri and colleagues report that clathrin plaques are mechanosensitive signaling platforms. By plating HeLa (but also Caco-2 and HepG2) cells on glass or polyacrylamide gels with different stiffness, they observed plaque formation in response to increasing substrate stiffness but independent of contractile forces and focal adhesion formation. The authors observed a strong co-localization of $\alpha\beta5$ integrin to clathrin plaques and show that plaque formation depends on the strong engagement of $\alpha\beta5$ integrin to its ECM ligand vitronectin. Interestingly, plaque formation was also observed with an independent, un-physiological receptor-ligand pair on stiff substrates because of frustrated endocytosis. Finally, they report that the plaque structures serve as signaling hubs on stiff substrates to promote Erk-dependent signaling and cell proliferation.

The manuscript is well written and convincingly argued. Some findings confirm already known data (e.g. that the plaque structures serve as signaling hubs) but the subject area and experimental findings are of interest to a general cell biology readership. In my view, some experiments require additional controls and the $\alpha\beta5$ integrin-dependency further mechanistic clarification.

Specific points:

- 1) Talin knockdown experiment, Figure 2: Previous publications (Zhang et al., NCB 2008; Theodosiou et al., Elife 2015) show that talin-1 and -2 depleted or knockout fibroblasts fail to adhere and spread or only spread for a brief period. It is surprising that HeLa cells adhered (and likely spread) in their experiments. Were talin-1 and -2 depleted? The siRNAs are not mentioned in the method section. How efficient is the talin knockdown and to what extent were the authors able to inhibit FA assembly?
- 2) It is interesting that MDA-MB-231 cells do not express $\alpha v\beta 5$ integrin and form only dynamic clathrin-coated pits but not plaques. Did the authors express $\alpha v\beta 5$ integrin in MDA-MB-231 cells to see if $\alpha v\beta 5$ integrin is sufficient to form plaques on stiff substrates?
- 3) It is a very interesting observation that clathrin plaque formation is dependent on $\alpha v\beta 5$ integrin. What is special about $\alpha v\beta 5$ integrin? Do the authors have an idea if the ligand (VN)-receptor interaction is special or are intracellular events distinct in the case of $\alpha v\beta 5$ integrin? Could other integrin receptors also induce plaque formation if the ligand would be covalently linked to the surface (such as FN and $\alpha 5\beta 1$ integrin)?
- 4) In line with the point above, it is commonly thought that integrins require the interaction with talin and kindlin to be able to bind their ligand and maintain this interaction. Are talin and kindlin recruited to the clathrin plaque to allow $\alpha v\beta 5$ integrin binding to vitronectin or is vitronectin binding independent of talin and kindlin?
- 5) Extended figure 6: Vitronectin coating did not result in more static CCSs on soft substrates. Is there a way so show that polyacrylamide gels are sufficiently coated with vitronectin?
- 6) Figure 4: Erk activation was mostly measured in cells under equilibrium condition, e.g. grown under standard culture conditions for several hours. Under these conditions, effects such as differences in the EGFR levels or intracellular trafficking of the EGFR could contribute to the observed effects. Are EGFR (surface) levels comparable between the tested conditions? Have the authors in addition analyzed Erk activation in after acute EGF stimulation in few time points (5', 15', 30')? One would perhaps expect a similar activation kinetics but a longer duration of Erk activation in cells with clathrin plaques.
- 7) Figure 4F: Why has CDC and AP-2 an effect under these conditions but not in Extended Figure 8? Is it because cells seeded on anti-mCherry-antibody coated surfaces do not form focal adhesions (did the authors check for this?) or because of the different readout (microscopy vs blot)?

Minor points:

- 1) Line 120f: The authors write that "large and long-lived CCSs were only detected at cell/ECM contact areas but not in non-adherent regions...". However, the quantification in extended figure 6G shows static CCSs in non-adherent regions.
- 2) Figure 2C: $\alpha v\beta 5$ integrin shows reduced co-localization with α -adaplin on soft substrates. Is $\alpha v\beta 5$ integrin more recruited into focal adhesions under these conditions?

Reviewer #3 (Remarks to the Author):

In this work, Bashieri and co-workers are correlating the presence of clathrin coated plaques with the mechano properties of the substrate. They report that on hard substrate, cells display classical endocytic clathrin coated pits (CCPs) and plaques while, on soft substrate, only CCPs are present. They show that the presence of plaques on the hard substrate is regulated by $\alpha v\beta 5$ and propose that the presence of plaques is the result of a frustrated endocytosis of the integrins. Additionally, they report that these plaques represent a signaling platform for EGF and Erk and that stiff substrates activate these signaling pathways inducing cell proliferation. They conclude by proposing that plaques are mechanotransduction structures that sense substrate rigidity to regulate cellular functions.

Although I believe that the work is of great interest not only in the field of clathrin but also in the field of cell migration and that it is elegantly performed and brings novel perspective to the functions of clathrin plaques (which is critically missing), some extra controls and experiments should be performed to support the conclusions and model of the authors.

In a nutshell, I found the part on the "signaling platform" very difficult to read and understand. This part should be detailed more, likely by generating several figures, but also by performing extra experiments that will definitively prove the proposed model of the authors. Additionally, I believe the terms mechanosensitive or mechanotransduction are somehow an overstatement. I fully agree with the authors that the mechanical properties of the substructure influence plaques formation. However, mechanotransduction, at least in the FA field, is defined by the fact that cells generate forces on the substrate and as such the stiffness of the substrate influences signal transduction. In the case of clathrin plaques, where is the force, is there any force? Since the authors show that it is acto-mysin independent, how can such structures apply force on the ECM?

My specific comments to the manuscript are:

Major comments:

- The authors define plaques in this work as CCS having a lifetime superior to 5 min. In the 0,1 kPa substrate, there is very little to no static CCS (plaques). Plaques are not only defined by their lifetime but also by their ultrastructural organization. The authors should show a detailed analysis of the lifetime and fluorescence profile of the CCS on 0,1 kPa substrate since the dynamic structures look different on the kymogram on fig 1B. Similarly, for the SEM, in extended figure 1B, the author should show the picture of the 0,1kPa or 5kPa gels to make it consistent with the rest of the data and to really prove that no plaques are present in soft substrate.
- MDA-MB-231 cells do not display plaques and the authors claim that this is due to the absence of $\alpha v \beta 5$ integrin. This is a very interesting observation. This suggests that MDA-MB-231 do not make plaque because they cannot bind vitronectin due to the absence of $\alpha v \beta 5$ integrin. The authors need to overexpress $\alpha v \beta 5$ in MDA-MB-231 to address whether over-expression induces the formation of plaques. This will strengthen their model considerably.
- The authors show that knock-down of αv or $\beta 5$ integrins results in a complete loss of plaques. They also show that pre-coating glass with vitronectin induces the formation of plaques. They conclude that plaques are the result of $\alpha v \beta 5$ binding vitronectin. As integrins αv and $\beta 5$ can assemble with other integrin sub-units to form different heterodimers, the authors should address whether their phenotype is strictly specific to $\alpha v \beta 5$ or similar if they knock-down $\alpha v \beta 3$ or $\alpha 5 \beta 1$. This is extremely relevant as $\alpha v \beta 3$ also binds vitronectin and the authors themselves have previously published that $\beta 1$ integrin is responsible for the formation of tubular plaque structures on collagen fibers. Additionally, coating of collagen or poly-L-lysine induces the formation of plaques, collagen recruits other integrins and adhesion to poly-L-lysine is integrin independent. I do not doubt the phenotype of the authors (that $\alpha v \beta 5$ is important) but we yet cannot conclude whether or not it is strictly $\alpha v \beta 5$ mediated.
- This comment goes in line with my previous one:
The authors propose that a plaque is the result of a frustrated endocytosis (clearly show with their TrfR-cherry approach) and suggest that in cells, plaques are regions where $\alpha v \beta 5$ cannot be internalized. However, they never show that $\alpha v \beta 5$ gets to be internalized when cilengitide is added to the cells. The authors should express tagged versions of $\alpha v \beta 5$ and look at internalization upon cilengitide treatment. Alternatively, the authors should knock-down integrin specific adaptors, like Dab2, Numb or ARH as these should result in the loss of plaques according to their model.
- The TrfR-cherry is a very nice experiment which shows a similar phenotype as observed for $\alpha v \beta 5$ integrin. However, it is not because there is a similar end-point phenotype that one can conclude that it is caused by the same molecular mechanism. Frustrated endocytosis is very likely

responsible for the plaque formation in the TrfR-cherry setup but, there is no evidence that this is the case for the $\alpha\text{v}\beta\text{5}$ integrin under physiological conditions. Importantly from fig3E: there is no correlation between the TrfR signal and the plaques. Can the authors explain this? From the figure, I will conclude that there is no relation between plaque formation and attachment to the TrfR. Importantly, The TrfR-mcherry experiments need to be performed on soft substrate. If the formation of clathrin plaques is driven by frustrated endocytosis, in this setup, plaques should be observed. If plaque formation is also driven by mechanical properties of the ECM, in this condition, despite the anti-cherry antibodies, plaques should not form. Or should we expect plaques because we remain in a frustrated configuration with the anti-cherry antibody. This will be a nice addition as the authors did not show whether $\alpha\text{v}\beta\text{5}$ are internalized or not.

- As mentioned above, I found the section on plaque as signaling platform very difficult to follow. Too many data and systems are presented in a single figure. Besides this style issue, the importance of integrin signaling in Erk activation appears to be completely ignored. In the $\alpha\text{v}\beta\text{5}$ knock-down cells, the authors show that the number of FAs is not affected, however, they should control that the activity is not decreased upon $\alpha\text{v}\beta\text{5}$ knock-down. Similarly, on soft substrates, there is less P-Erk, but what about FA, because on soft substrates the cell will have very few FAs (as such activating less Erk).

- In Figure 4EF: I don't understand the rationale. The plaques that are induced by the Trf-cherry approach are fundamentally different from the natural ones and are likely not going to be functionally active for Erk activation. As a matter of fact, can the authors clarify what activates Erk in this system? I believe these experiments will be easier to conclude if EGF was used.

-The proliferation part is a nice addition to the paper as it starts to provide some functions to plaques. The authors show that plaques induced by integrin act as signaling platforms stimulating Erk signaling and inducing cell proliferation. To fully prove this model, and to uncouple this phenotype from integrin, the authors could once more exploit their TrfR-cherry system and rescue $\alpha\text{v}\beta\text{5}$ knock-down cells by generating frustrated endocytic plaques. This should serve as signaling platform and rescue proliferation.

Minor comments:

-Page 3: " Although plaques have been widely described and shown to be enriched in signaling receptors and integrins" some references should be added.

- Can the author make a short statement to illustrate what the young's modulus mean in a biological context? e.g 0,1 kPa would be brain and 32kPa will be bone?

- The authors should show that knock-down of talin was indeed efficient. Either by a western blot of talin or by immunofluorescence staining of FA.

-The drop of association between AP2 and $\alpha\text{v}\beta\text{5}$ integrin (pearson correlation) on soft substrate is mostly due to the fact that plaques are less present. The authors should measure the intensity ratio of $\alpha\text{v}\beta\text{5}$ integrin and adaptin on the different substrate (like in Fig 2B) but maybe there is not enough plaque left to be analyzed.

- Page 5: "inhibition of either", I guess the author mean knock down?

- The experiments with cilengitide looks more convincing and are more specific than the trypsin ones. I suggest to put the cilengitide picture in the main text and trypsin in supplementary.

Reviewers' comments:

Answers are in blue.

Reviewer #1 (Remarks to the Author):

The manuscript by Baschieri et al. demonstrates that flat clathrin-coated structures (CCSs) named plaques are contractility-independent mechanosensitive transduction hubs. In this regard, the Authors found that plaques generate upon substrate rigidity independent of actin and myosin-II activity. In particular, plaques originate from the elasticity-regulated affinity of $\alpha v \beta 5$ integrin for the ECM and serve as signaling platforms toward Erk activation and cell proliferation on stiff environments.

The matter is interesting and deserve further research roads and applications. However, I believe that it could be difficult for a general audience to follow the consecutive meaning of the experimental design as the Authors have written the manuscript. Moreover, many figures are shown in black and white, the readers would better appreciate the findings if shown in color.

We thank the reviewer for his comments. We have reorganized the section of the manuscript describing the role of plaques in regulating signaling pathways as the other reviewers also pointed to a lack of clarity regarding this part. Regarding the color issue, it is generally accepted in the cell biology field that gray levels pictures are easier to read by human eyes. Some pictures are shown in color where it brings a relevant information.

Comments

The Authors ascertained that certain cells as HeLa, HepG2 and Caco2 cells display both flat clathrin structures called plaques and dynamic clathrin-coated pits, whereas MDA-MB-231 cells only the latter. The Authors no longer observed plaques in $\alpha v \beta 5$ depleted cells, hence it could be interesting to engineer MDA-MB-231 cells (and other cells displaying similar features as MDA-MB-231 cells) to express $\alpha v \beta 5$ in order to strengthen the results observed on the role of $\alpha v \beta 5$. In addition, if these results may be linked to the aggressive and metastatic behavior of MDA-MB-231 (and other cells), the data presented may be of particular interest toward the role of plaques in cancer progression.

This point was also raised by other reviewers and we now provide new figures showing that 1) αv and $\beta 5$ integrins levels are very low in MDA-MB-231 cells as compared to HeLa cells (Supplementary Fig. 4h). Overexpressing αv and $\beta 5$ integrins in MDA-MB-231 cells leads to the accumulation of long-lived CCSs, as observed in control HeLa cells (Supplementary Fig. 4i).

In addition, we now provide more direct evidence for a role of plaques in regulating cell proliferation (Fig. 6g-i), which we believe is the main function of plaques.

The Authors suggested an intriguing chance regarding plaques as promiscuous platforms for different growth factor-activated signaling, as plaques were strongly marked with an anti-phosphotyrosine antibody upon EGF and HGF stimulation. Thereafter, the Authors found that plaques regulate Erk signaling independently of $\beta 5$ -integrin that is required for plaque formation and therefore EGFR-mediated signaling. The Authors could explain this result that appears contrasting with the classical transduction pathway linking EGFR with Erk signaling. In this regard, it could be interesting to

evaluate upon EGF stimulation both Erk activation and the expression levels of c-fos, which is a well acknowledged molecular sensor of this signaling and closely related to cell proliferation.

We apologize for the lack of clarity in the original manuscript regarding our model. We propose that while $\alpha v\beta 5$ regulates the formation of plaques, this integrin does not directly control Erk signaling at these structures. Rather, signaling receptors (including the tested RTKs but presumably also many others) recruitment at plaques is required for sustained Erk signaling. This is supported by our new data showing that EGFR and HGFR are still recruited at artificial plaques (TfR-mCherry/anti-mCherry antibody system; Fig. 5e-g) in the absence of the integrin and that in these conditions, Erk activity is restored (Fig 4e, f).

Reviewer #2 (Remarks to the Author):

Cells internalize membrane proteins through the formation of clathrin-coated structures that can appear as dynamic clathrin-coated pits or long-lived structures called, clathrin lattices or plaques. These clathrin lattices have been suggested to be sites of adhesion and may act as signaling platforms, which concentrate activated signaling receptors including receptor tyrosine kinases (RTKs). The manuscript by Baschieri and colleagues report that clathrin plaques are mechanosensitive signaling platforms. By plating HeLa (but also Caco-2 and HepG2) cells on glass or polyacrylamide gels with different stiffness, they observed plaque formation in response to increasing substrate stiffness but independent of contractile forces and focal adhesion formation. The authors observed a strong co-localization of $\alpha\beta5$ integrin to clathrin plaques and show that plaque formation depends on the strong engagement of $\alpha\beta5$ integrin to its ECM ligand vitronectin. Interestingly, plaque formation was also observed with an independent, un-physiological receptor-ligand pair on stiff substrates because of frustrated endocytosis. Finally, they report that the plaque structures serve as signaling hubs on stiff substrates to promote Erk-dependent signaling and cell proliferation.

The manuscript is well written and convincingly argued. Some findings confirm already known data (e.g. that the plaque structures serve as signaling hubs) but the subject area and experimental findings are of interest to a general cell biology readership. In my view, some experiments require additional controls and the $\alpha\beta5$ integrin-dependency further mechanistic clarification.

Specific points:

1) Talin knockdown experiment, Figure 2: Previous publications (Zhang et al., NCB 2008; Theodosiou et al., Elife 2015) show that talin-1 and -2 depleted or knockout fibroblasts fail to adhere and spread or only spread for a brief period. It is surprising that HeLa cells adhered (and likely spread) in their experiments. Were talin-1 and -2 depleted? The siRNAs are not mentioned in the method section. How efficient is the talin knockdown and to what extent were the authors able to inhibit FA assembly?

We apologize for the omission. Our siRNA only targets Talin1 which is the main Talin isoform. This is now stated in the material and methods section. We now provide evidence showing that Talin1 staining disappears (Supplementary Fig. 2i) and that FAs are strongly reduced in size and number (Supplementary Fig. 2j) in Talin-siRNA-treated cells.

2) It is interesting that MDA-MB-231 cells do not express $\alpha\beta5$ integrin and form only dynamic clathrin-coated pits but not plaques. Did the authors express $\alpha\beta5$ integrin in MDA-MB-231 cells to see if $\alpha\beta5$ integrin is sufficient to form plaques on stiff substrates?

This is a very good point also raised by the two other referees. We now provide new figures showing that 1) αv and $\beta 5$ integrin levels are very low in MDA-MB-231 cells as compared to HeLa cells (Supplementary Fig. 4h) and 2) overexpressing αv and $\beta 5$ integrins in MDA-MB-231 cells leads to the formation of long-lived CCSs, as observed in control HeLa cells (Supplementary Fig. 4i). These new pieces of evidence reinforce our conclusion that $\alpha\beta5$ is instrumental for plaque formation.

3) It is a very interesting observation that clathrin plaque formation is dependent on $\alpha\text{v}\beta\text{5}$ integrin. What is special about $\alpha\text{v}\beta\text{5}$ integrin? Do the authors have an idea if the ligand (VN)-receptor interaction is special or are intracellular events distinct in the case of $\alpha\text{v}\beta\text{5}$ integrin? Could other integrin receptors also induce plaque formation if the ligand would be covalently linked to the surface (such as FN and $\alpha\text{5}\beta\text{1}$ integrin)?

We show that β1 or β3 depletion does not prevent plaque formation (Supplementary Fig. 4e, f) pointing to a special role for $\alpha\text{v}\beta\text{5}$ in this process. We believe that the key point for the formation of plaques is the affinity of the integrin for the clathrin machinery. β5 has been reported to have a strong affinity for Dab2 and Numb, two well-known clathrin-adaptors (Calderwood DA et al, PNAS 2003, DOI: 10.1073/pnas.262791999) while β1 only had a weak affinity for Numb in this report. Immobilizing the TfR, that has a strong affinity for AP-2, leads to CCSs frustration and plaque formation. Thus, we believe that immobilizing any receptor that has a strong enough affinity for the clathrin machinery may result in plaque formation.

4) In line with the point above, it is commonly thought that integrins require the interaction with talin and kindlin to be able to bind their ligand and maintain this interaction. Are talin and kindlin recruited to the clathrin plaque to allow $\alpha\text{v}\beta\text{5}$ integrin binding to vitronectin or is vitronectin binding independent of talin and kindlin?

Recruitment of integrins into CCSs usually relies on the Phospho Tyrosine Binding (PTB) domain-containing proteins Dab2 and/or Numb that are also well-described clathrin-adaptors. We now provide evidence that both Dab2 and Numb are required to form fully developed clathrin-coated plaques (Supplementary Fig. 5). On the opposite, Talin was not found associated with CCSs (Supplementary Fig. 2i).

5) Extended figure 6: Vitronectin coating did not result in more static CCSs on soft substrates. Is there a way so show that polyacrylamide gels are sufficiently coated with vitronectin?

We observed that cells efficiently spread on vitronectin-coated polyacrylamide gels which is not the case when gels are not coated (data not shown). This suggests that gels are sufficiently coated with vitronectin.

6) Figure 4: Erk activation was mostly measured in cells under equilibrium condition, e.g. grown under standard culture conditions for several hours. Under these conditions, effects such as differences in the EGFR levels or intracellular trafficking of the EGFR could contribute to the observed effects. Are EGFR (surface) levels comparable between the tested conditions? Have the authors in addition analyzed Erk activation in after acute EGF stimulation in few time points (5', 15', 30')? One would perhaps expect a similar activation kinetics but a longer duration of Erk activation in cells with clathrin plaques.

We now report that surface levels of EGFR are not significantly modulated in the absence of plaques ($\beta 5$ siRNAs; Supplementary Fig. 11a). In addition, we provide new data showing that the strength of Erk activation following acute EGF stimulation is reduced in the absence of plaques (Fig. 5c, d). Our model is that signaling by any receptor (upon acute activation or steady-state activation in the presence of serum) is potentiated when activated receptors accumulate at plaques.

7) Figure 4F: Why has CDC and AP-2 an effect under these conditions but not in Extended Figure 8? Is it because cells seeded on anti-mCherry-antibody coated surfaces do not form focal adhesions (did the authors check for this?) or because of the different readout (microscopy vs blot)?

We agree with the referee that this point was not clear in the original version of the manuscript. We now provide several pieces of evidence pointing at the existence of a crosstalk between clathrin-coated plaques and FAs. Indeed, when CCSs (and thus plaques) formation is inhibited by knocking down the AP2 complex or clathrin, $\alpha \beta 5$ is targeted at FAs while it is only slightly present at these structures in control cells (Supplementary Fig. 5f, and Supplementary Fig. 6a). This leads to FAs enlargement (Supplementary Fig. 6f, g and Supplementary Fig. 9b) and an increased signaling activity of FAs (as denoted by phosphotyrosine staining; Supplementary Fig. 9b, d) in a $\beta 5$ -integrin-dependent manner (Supplementary Fig. 6f, g). We believe this is the reason why CHC or AP-2 depletion did not show any effect on Erk activity in cells cultured on glass. Indeed, in these experimental conditions, inhibiting both FAs formation and CHC or AP-2 resulted in a strong reduction in Erk activity while FAs inhibition alone did not (Supplementary Fig. 9e, f).

In the case of experiments performed with the anti-mCherry antibody, the antibody is anchored to poly-L-lysine-coated glass. Poly-L-lysine poorly engages integrins and consequently, we measured fewer and smaller FAs in these conditions (Supplementary Fig. 10a-c). This explains why CHC and AP-2 did not have the same effect in the two experimental conditions.

Minor points:

1) Line 120f: The authors write that “large and long-lived CCSs were only detected at cell/ECM contact areas but not in non-adherent regions...”. However, the quantification in extended figure 6G shows static CCSs in non-adherent regions.

The referee is right. We changed the sentence in the main text for “most large and long-lived CCSs were detected at cell/ECM contact areas rather than at non-adherent regions of the ventral plasma membrane”.

2) Figure 2C: $\alpha \beta 5$ integrin shows reduced co-localization with α -adapatin on soft substrates. Is $\alpha \beta 5$ integrin more recruited into focal adhesions under these conditions?

We observed an almost complete loss of focal adhesions on soft substrates (5 and 0.1 kPa; data not shown). Thus it is very difficult to accurately measure the colocalisation between $\alpha \beta 5$ integrin and Focal adhesions markers in these conditions. In any case, the remaining $\alpha \beta 5$ staining does not accumulate at the periphery of cells seeded on soft substrate arguing against a specific enrichment at Focal adhesion in these conditions (Supplementary Fig.3).

Reviewer #3 (Remarks to the Author):

In this work, Bashieri and co-workers are correlating the presence of clathrin coated plaques with the mechano properties of the substrate. They report that on hard substrate, cells display classical endocytic clathrin coated pits (CCPs) and plaques while, on soft substrate, only CCPs are present. They show that the presence of plaques on the hard substrate is regulated by $\alpha v\beta 5$ and propose that the presence of plaques is the result of a frustrated endocytosis of the integrins. Additionally, they report that these plaques represent a signaling platform for EGF and Erk and that stiff substrates activate these signaling pathways inducing cell proliferation. They conclude by proposing that plaques are mechanotransduction structures that sense substrate rigidity to regulate cellular functions.

Although I believe that the work is of great interest not only in the field of clathrin but also in the field of cell migration and that it is elegantly performed and brings novel perspective to the functions of clathrin plaques (which is critically missing), some extra controls and experiments should be performed to support the conclusions and model of the authors.

In a nutshell, I found the part on the “signaling platform” very difficult to read and understand. This part should be detailed more, likely by generating several figures, but also by performing extra experiments that will definitively prove the proposed model of the authors.

We thank the reviewer for raising this point. We reorganized the section describing the role of plaques in signaling and we now provide three main figures to clarify our conclusions on that matter (Fig 4, 5 and 6).

Additionally, I believe the terms mechanosensitive or mechanotransduction are somehow an overstatement. I fully agree with the authors that the mechano properties of the substructure influence plaques formation. However, mechanotransduction, at least in the FA field, is defined by the fact that cells generate forces on the substrate and as such the stiffness of the substrate influences signal transduction. In the case of clathrin plaques, where is the force, is there any force? Since the authors show that it is acto-mysin independent, how can such structures apply force on the ECM?

We agree with the referee that it is generally accepted that mechanotransduction mechanisms usually implies that some forces are exerted by the cell on its environment. This is the reason why we tested a role for cell contractility before to conclude that it does not play a role in plaque formation (Fig. 1 d, e, f) nor in plaque-regulated signaling (Fig. 6a, b, e). We cannot rule out that some forces are exerted at plaques. However, in silico models suggested that integrin cluster affinity for the substrate could be modulated by substratum stiffness in the absence of forces (Qian J et al, PLoS One 2010; Du, J et al PNAS 2011). Although, we believe that those models need to be confirmed in cellulo, we also believe it is out of the scope of this paper to investigate this possibility.

My specific comments to the manuscript are:

Major comments:

- The authors define plaques in this work as CCS having a lifetime superior to 5 min. In the 0,1 kPa substrate, there is very little to no static CCS (plaques). Plaques are not only defined by their lifetime but also by their ultrastructural organization. The authors should show a detailed analysis of the lifetime and fluorescence profile of the CCS on 0,1 kPa substrate since the dynamic structures look different on the kymogram on fig 1B. Similarly, for the SEM, in extended figure 1B, the author should show the picture of the 0,1kPa or 5kPa gels to make it consistent with the rest of the data and to really prove that no plaques are present in soft substrate.

We now provide a more precise measurement of the lifetime of CCSs in the different conditions of rigidity (Supplementary Fig. 2a). We also performed SEM analysis of CCSs on 0.1 and 5 kPa gels and confirmed that no plaques are detected at the ultrastructural levels in these conditions (Supplementary Fig. 1b).

- MDA-MB-231 cells do not display plaques and the authors claim that this is due to the absence of $\alpha v\beta 5$ integrin. This is a very interesting observation. This suggest that MDA-MB-231 do not make plaque because they cannot bind vitronectin due to the absence of $\alpha v\beta 5$ integrin. The authors need to overexpress $\alpha v\beta 5$ in MDA-MB-231 to address whether over-expression induce the formation of plaques. This will strengthen their model considerably.

This is a very good point also raised by the two other referees. We now provide new figures showing that 1) αv and $\beta 5$ integrins levels are very low in MDA-MB-231 cells as compared to HeLa cells (Supplementary Fig. 4h) and 2) overexpressing αv and $\beta 5$ integrins in MDA-MB-231 cells leads to the formation of long-lived CCSs, as observed in control HeLa cells (Supplementary Fig. 4i). These new pieces of evidence reinforce our conclusion that $\alpha v\beta 5$ is instrumental for plaque formation.

- The authors show that knock-down of αv or $\beta 5$ integrins results in a complete loss of plaques. They also show that pre-coating glass with vitronectin induces the formation of plaques. They conclude that plaques are the result of $\alpha v\beta 5$ binding vitronectin. As integrins αv and $\beta 5$ can assemble with other integrin sub-units to form different heterodimers, the authors should address whether their phenotype is strictly specific to $\alpha v\beta 5$ or similar if they knock-down $\alpha v\beta 3$ or $\alpha 5\beta 1$. This is extremely relevant as $\alpha v\beta 3$ also binds vitronectin and the authors themselves have previously published that $\beta 1$ integrin is responsible for the formation of tubular plaques structures on collagen fibers.

We now report that $\beta 1$ or $\beta 3$ depletion does not prevent plaque formation (Supplementary Fig. 4e, f) pointing to a specific role for $\alpha v\beta 5$ in this process.

Additionally, coating of collagen or poly-L-lysine induces the formation of plaques, collagen recruits other integrins and adhesion to poly-L-lysine is integrin independent. I do not doubt the phenotype of the authors (that $\alpha v\beta 5$ is important) but we yet cannot conclude whether or not it is strictly $\alpha v\beta 5$ mediated.

Vitronectin is abundant in the serum and non-specifically adsorbs to the glass during cell plating (Hayman et al., *Exp Cell Res.* 1985 Oct;160(2):245-58). When coverslips are coated with collagen, vitronectin from the serum could still adsorb on the glass, or on collagen itself as it was shown that vitronectin binds to collagen (Gebb et al., *J Biol Chem.* 1986 Dec 15;261(35):16698-703).

The referee is right that poly-L-lysine does not engage integrins but it is likely that ECM proteins found in the serum (present in these experiments) bind to poly-L-lysine or even directly to the glass in between poly-L-lysine.

- This comments goes in line with my previous one:

The authors propose that a plaque is the result of a frustrated endocytosis (clearly show with their TrfR-cherry approach) and suggest that in cells, plaques are regions where $\alpha v\beta 5$ cannot be internalized. However, they never show that $\alpha v\beta 5$ gets to be internalized when cilengitide is added to the cells. The authors should expressed tagged versions of $\alpha v\beta 5$ and look at internalization upon cilengitide treatment. Alternatively, the authors should knock-down integrin specific adaptors, like Dab2, Numb or ARH as these should result in the loss of plaques according to their model.

This is a very good point. Following the referee's advices, we expressed GFP-tagged $\beta 5$ and showed that it is not present in TrfR-positive endosomes in control cells (Supplementary Fig. 8d, upper panels). However, GFP- $\beta 5$ colocalized with the TrfR in endosomes shortly after cilengitide treatment, demonstrating that the integrin has been internalized (Supplementary Fig. 8d, lower panels).

We also tested for the role of the integrin and clathrin adaptors Dab2 and Numb and we now report that both adaptors are required for the formation of fully developed clathrin-coated plaques (Supplementary Fig. 5).

As suggested by the referee, these two pieces of evidence reinforce our model that $\alpha v\beta 5$ control CCSs frustration and that this process involves the $\beta 5$ -binding clathrin adaptors Dab2 and Numb (Calderwood et al, *PNAS* 2003).

- The TrfR-cherry is a very nice experiment which shows a similar phenotype as observed for $\alpha v\beta 5$ integrin. However, it is not because there is a similar end-point phenotype that one can conclude that it is caused by the same molecular mechanism. Frustrated endocytosis is very likely responsible for the plaque formation in the TrfR-cherry setup but, there is no evidence that this is the case for the $\alpha v\beta 5$ integrin under physiological conditions. Importantly from fig3E: there is no correlation between the TrfR signal and the plaques. Can the authors explain this? From the figure, I will conclude that there is no relation between plaque formation and attachment to the TrfR. Importantly, The TrfR-mcherry experiments need to be performed on soft substrate. If the formation of clathrin plaques is driven by frustrated endocytosis, in this setup, plaques should be observed. If plaque formation is also driven by mechanical properties of the ECM, in this condition, despite the anti-cherry antibodies, plaques should not form. Or should we expect plaques because we remain in

a frustrated configuration with the anti-cherry antibody. This will be a nice addition as the authors did not show whether avb5 are internalized or not.

We apologize for the missing information regarding the localization of cell surface TfR-mcherry at CCSs. The apparent lack of colocalization in the original Fig. 3E is due to the fact that TfR is also (and mostly) present in endosomes and vesicles that are very bright and that mask the cell surface signal of the receptor. We now provide TIRF images showing that TfR-mCherry is indeed present at CCSs (Supplementary Fig. 8e).

We also performed the TfR-mCherry experiments on soft gels and we now report that immobilizing the TfR in these conditions also leads to the accumulation of long-lived CCSs, although to a lesser extent as compared to the glass condition (Supplementary Fig. 8f). Together with our new findings that $\alpha v \beta 5$ is internalized upon ciliengitide treatment, we conclude that plaque formation is indeed a consequence of endocytosis frustration.

- As mentioned above, I found the section on plaque as signaling platform very difficult to follow. Too many data and systems are presented in a single figure. Besides this style issue, the importance of integrin signaling in Erk activation appears to be completely ignored. In the $\alpha v \beta 5$ knock-down cells, the authors show that the number of FAs is not affected, however, they should control that the activity is not decreased upon $\alpha v \beta 5$ knock-down. Similarly, on soft substrates, there is less P-Erk, but what about FA, because on soft substrates the cell will have very few FAs (as such activating less Erk).
- In Figure 4EF: I don't understand the rationale. The plaques that are induced by the Trf-cherry approach are fundamentally different from the natural ones and are likely not going to be functionally active for Erk activation. As a matter of fact, can the authors clarify what activates Erk in this system? I believe these experiments will be easier to conclude if EGF was used.

We believe these two points are similar and derive from the lack of clarity of our exposed model in the initial submission, for which we apologize.

First, following the reviewer's advice, we looked at phospho-FAK levels upon $\beta 5$ depletion and concluded that $\alpha v \beta 5$ does not regulate FAK activity in these conditions (Supplementary Fig. 6d, e). This may not be surprising because $\alpha v \beta 5$ is mostly present at plaques and only a minor fraction could be seen at FAs in control conditions (Supplementary Fig. 5f, Supplementary Fig. 6a). As a matter of fact, integrin $\beta 5$ depletion does not affect size and number of focal adhesions (Supplementary Fig. 6b, c, g).

Second, we now report that when FAs formation is inhibited (Talin siRNA) there is at best only a minor effect on Erk activity (Supplementary Fig. 9e, f). Accordingly, blebbistatin treatment, which inhibits contractility and hence FAs maturation, did not significantly modulate P-Erk levels (Fig. 6e, f). This suggests that, at least in HeLa cells and in our particular experimental conditions, FAs do not strongly regulate steady-state Erk activity.

However, we show that clathrin-coated plaques strongly regulates steady-state Erk activity (Fig 4a, b, e, f and Fig. 6e, f) as well as EGF-induced Erk activation (Fig. 5c, d). Our model is that $\alpha v\beta 5$ -itself does not directly control Erk signaling at plaques. We rather propose that $\alpha v\beta 5$ is required to assemble plaques and that, once assembled, plaques serve as signaling platform for different receptors, including (but presumably not limited to) the two RTKs we tested (EGFR and HGFR). This is supported by the fact that, in the absence of $\alpha v\beta 5$, artificial induction of plaques formation with the anti-mCherry antibody restores Erk activity (Fig. 4e, f). Importantly, these artificial structures are still able to recruit RTKs (Fig. 5e, f) and are strongly stained by an anti-phosphotyrosine antibody upon stimulation (Fig 5g), similar to genuine plaques.

-The proliferation part is a nice addition to the paper as it starts to provide some functions to plaques.

The authors show that plaques induced by integrin act as signaling platforms stimulating Erk signaling and inducing cell proliferation. To fully prove this model, and to uncouple this phenotype from integrin, the authors could once more exploit their TfrR-cherry system and rescue $\alpha v\beta 5$ knock-down cells by generating frustrated endocytic plaques. This should serve as signaling platform and rescue proliferation.

We thank the referee for this nice suggestion. We now report that inducing artificial plaques using the TfrR-mCherry system boosts cell proliferation in cells without $\alpha v\beta 5$ (Fig. 6i). This reinforces our conclusion that these artificial plaques are functional and that Erk activation and cell proliferation are not strictly dependent on the integrin itself but rather on the presence of plaques.

Minor comments:

-Page 3:“ Although plaques have been widely described and shown to be enriched in signaling receptors and integrins” some references should be added.

References have been added in the main text.

- Can the author make a short statement to illustrate what the young's modulus mean in a biological context? e.g 0,1 kPa would be brain and 32kPa will be bone?

A statement has been added in the introduction section of the revised manuscript.

- The authors should show that knock-down of talin was indeed efficient. Either by a western blot of talin or by immunofluorescence staining of FA.

Supplementary Fig. 2i now shows that Talin knockdown was efficient.

-The drop of association between AP2 and $\alpha v\beta 5$ integrin (pearson correlation) on soft substrate is mostly due to the fact that plaques are less present. The authors should measure the intensity ratio of $\alpha v\beta 5$ integrin and adaptin on the different substrate (like in Fig 2B) but maybe there is not enough plaque left to be analyzed.

Indeed, it is difficult to provide such a measurement on soft substrate because there is essentially no more plaques and the data may consequently not be relevant.

- Page 5: "inhibition of either", I guess the author mean knock down?

The text has been modified according to the referee's suggestion.

- The experiments with cilengitide looks more convincing and are more specific than the trypsin ones. I suggest to put the cilengitide picture in the main text and trypsin in supplementary.

We decided to keep the trypsin experiment in the main figure because the cilengitide experiment is already there anyway (Fig. 3d) and there is enough space for both.

REVIEWERS' COMMENTS:

Reviewer #1 (Remarks to the Author):

As required, the Authors incorporated additional important experiments in the revised manuscript. Although the Authors did not deal with certain suggestions (i.e. gene expression changes linking ERK activation and proliferation), a very good work was done. The results are convincing and interesting, therefore these data would open new avenues in the specific field.

--

Reviewer #2 (Remarks to the Author):

The authors report that clathrin-coated structures are contractility-independent signaling hubs. They show that CCS formation depends on α v β 5 integrin and is a consequence of frustrated endocytosis. Most of my comments/questions during the revision have been addressed and I can support publication of this study in Nature Communication with this data included.

Few minor points that came up during the revision should be addressed.

Minor points:

- 1) The authors now specify in the method section that siRNAs against Talin-1 have been used. I would also suggest making this distinction in the text and Supplementary Figure 2 and writing about siTalin-1 instead of siTalin.
- 2) Line 96: Since FA assembly is not completely inhibited (Supplementary Figure 2j) I would rather write about "interfering with FA assembly".
- 3) Supplementary Figure 4e – Itgb1 and Itgb3 depletion. Did the author test the amount of the respective integrin subunits on the cell surface? From the western blot provided it seems that β 1 integrin siRNA treatment led to reduced levels of the low MW integrin form (which represents β 1 integrin in the endoplasmic reticulum) while the fully glycosylated, mature β 1 integrin (MW 125 kDa; representing the cell surface integrin) is still present at similar levels.
- 4) Re-expressing α v β 5 integrin in MDA-MB-231 enables the cells to form long-lived CCSs – however with a lower percentage compared to other cell lines including HeLa. Out of curiosity: Did the author determine and compare the amount of α v β 5 integrin between the cell lines to check if the fewer CCSs are the result of reduced α v β 5 integrin levels?
- 5) Line 217: It is written that "AP-2 or CHC knockdown induced a strong accumulation of phosphotyrosines at the enlarged FAs, in a β 5-integrin-dependent manner (Supplementary Fig. 9b-d)." However, the β 5-integrin-dependency is not shown in Supplementary Figure 9 but Supplementary Figure 6.

--

Reviewer #3 (Remarks to the Author):

The authors have perfectly responded to my comments and suggestions.

I recommend publication. I would like to congratulate the authors for this great work and exciting new findings concerning clathrin plaques. I am looking forward their next story on flat clathrin lattices.

There are very minor corrections that should be done before publication, see below:

Main text:

Lines 48-49: "CCSs can also serve as integrin-dependent adhesion structures ", I am not aware of this, do the authors have a reference clearly demonstrating this?

Line 209: beta5 knockdown would be more correct than "beta5-inhibition" to not mix it up with the cilengitide treatment

Line 247 and Figure 5a: The arrows on the CCPs give the impression that they are the most important CCSs although we should just compare them to plaques. Either pointing with another symbol to the plaques or removing the arrowheads would help the reader.

Line 288-297 (Discussion): Very short discussion and their findings are not put in the context of the literature. I know there is space limit, but maybe, if the editorial board accept it, the authors could have an extra page where they could discuss their findings in the context of papers related to clathrin-coated plaques as signaling platform or adhesive structure.

Methods:

Line 338-339: For what experiment were the Akt and pAkt antibody used?

Line 352-353: Catalogue numbers are missing for EGF, HGF, Blebbistatin, Gefitinib, Lantrunculin A and Cytochalasin D

Figures and legends:

Line 703, 717, 746: "Latter" should be later

Fig 4e: Please include scale for color-coding. Scale bar is missing in the images.

Fig 6f: Y-axis is too short, the last error bar is cut.

Fig 6g and h: Symbols cannot be distinguished they are too close to each other. Might be easier to read if they would be bigger or different color.

Supplementary:

Supp Fig 1a: Scale bar is missing in the images.

Supp Fig 2 legend: The last panel is mislabeled. It should be j instead of h

Supp Fig 8d: Scale bar is missing in the images.

Supp Fig 9d: "Tfluorescence" does this mean total fluorescence?

Supp Fig 11b: Scale bar is missing in the images.

General:

The authors should check for consistency of the use of hyphens for "x-coated" and "genome-edited"

POINT-BY-POINT RESPONSE TO REVIEWERS' COMMENTS:

Reviewer #1 (Remarks to the Author):

As required, the Authors incorporated additional important experiments in the revised manuscript. Although the Authors did not deal with certain suggestions (i.e. gene expression changes linking ERK activation and proliferation), a very good work was done. The results are convincing and interesting, therefore these data would open new avenues in the specific field.

--

Reviewer #2 (Remarks to the Author):

The authors report that clathrin-coated structures are contractility-independent signaling hubs. They show that CCS formation depends on alphaVbeta5 integrin and is a consequence of frustrated endocytosis. Most of my comments/questions during the revision have been addressed and I can support publication of this study in Nature Communication with this data included.

Few minor points that came up during the revision should be addressed.

Minor points:

1) The authors now specify in the method section that siRNAs against Talin-1 have been used. I would also suggest making this distinction in the text and Supplementary Figure 2 and writing about siTalin-1 instead of siTalin.

OK

2) Line 96: Since FA assembly is not completely inhibited (Supplementary Figure 2j) I would rather write about "interfering with FA assembly".

OK

3) Supplementary Figure 4e – Itgb1 and Itgb3 depletion. Did the author test the amount of the respective integrin subunits on the cell surface? From the western blot provided it seems that $\beta 1$ integrin siRNA treatment led to reduced levels of the low MW integrin form (which represents $\beta 1$

integrin in the endoplasmic reticulum) while the fully glycosylated, mature $\beta 1$ integrin (MW 125 kDa; representing the cell surface integrin) is still present at similar levels.

We didn't perform any surface staining of ITGB1. However we incubated HeLa cells with blocking antibodies for ITGB1 and we obtained the same results as for ITGB1 depletion (data not included in the manuscript).

4) Re-expressing $\alpha\beta 5$ integrin in MDA-MB-231 enables the cells to form long-lived CCSs – however with a lower percentage compared to other cell lines including HeLa. Out of curiosity: Did the author determine and compare the amount of $\alpha\beta 5$ integrin between the cell lines to check if the fewer CCSs are the result of reduced $\alpha\beta 5$ integrin levels?

Although this is an interesting point, we did not perform this analysis. It is also possible that the expression level of other players involved in plaque formation (like Dab2 or Numb) may vary between cell lines.

5) Line 217: It is written that “AP-2 or CHC knockdown induced a strong accumulation of phosphotyrosines at the enlarged FAs, in a $\beta 5$ -integrin-dependent manner (Supplementary Fig. 9b-d).” However, the $\beta 5$ -integrin-dependency is not shown in Supplementary Figure 9 but Supplementary Figure 6.

OK

--

Reviewer #3 (Remarks to the Author):

The authors have perfectly responded to my comments and suggestions.

I recommend publication. I would like to congratulate the authors for this great work and exciting new findings concerning clathrin plaques. I am looking forward their next story on flat clathrin lattices.

There are very minor corrections that should be done before publication, see below:

Main text:

Lines 48-49: “CCSs can also serve as integrin-dependent adhesion structures “, I am not aware of this, do the authors have a reference clearly demonstrating this?

We apologize for this mistake. The work describing that CCSs can serve as adhesive structures is now cited in the revised version of the manuscript.

Line 209: beta5 knockdown would be more correct than “beta5-inhibition” to not mix it up with the cilengitide treatment

OK

Line 247 and Figure 5a: The arrows on the CCPs give the impression that they are the most important CCSs although we should just compare them to plaques. Either pointing with another symbol to the plaques or removing the arrowheads would help the reader.

One symbol pointing to plaques was added.

Line 288-297 (Discussion): Very short discussion and their findings are not put in the context of the literature. I know there is space limit, but maybe, if the editorial board accept it, the authors could have an extra page where they could discuss their findings in the context of papers related to clathrin-coated plaques as signaling platform or adhesive structure.

No editorial request regarding this point.

Methods:

Line 338-339: For what experiment were the Akt and pAkt antibody used?

We apologize for the mistake. The description of these antibodies have been removed.

Line 352-353: Catalogue numbers are missing for EGF, HGF, Blebbistatin, Gefitinib, Lantrunculin A and Cytochalasin D

Catalogue numbers have been added.

Figures and legends:

Line 703, 717, 746: “Latter” should be later

OK

Fig 4e: Please include scale for color-coding. Scale bar is missing in the images.

OK

Fig 6f: Y-axis is too short, the last error bar is cut.

Corrected.

Fig 6g and h: Symbols cannot be distinguished they are too close to each other. Might be easier to read if they would be bigger or different color.

Colors have been added.

Supplementary:

Supp Fig 1a: Scale bar is missing in the images.

OK

Supp Fig 2 legend: The last panel is mislabeled. It should be j instead of h

OK

Supp Fig 8d: Scale bar is missing in the images.

OK

Supp Fig 9d: "Tfluorescence" does this mean total fluorescence?

Typo corrected.

Supp Fig 11b: Scale bar is missing in the images.

OK

General:

The authors should check for consistency of the use of hyphens for "x-coated" and "genome-edited"

OK